# Functional characteristics and computational model of abundant hyperactive loci in the human genome

Sanjarbek Hudaiberdiev*, Ivan Ovcharenko*

National Institute for Biotechnology and Information, National Library of Medicine, National Institutes of Health, Bethesda, United States

## eLife Assessment

This **valuable** study explores the sequence characteristics and conservation of high-occupancy target loci, regions in the human genome such as promoters and enhancers that are bound by a multitude of transcription factors. The computational analyses presented in this study are **solid**. This study would be a helpful resource for researchers performing ChIP-seq based analyses of transcription factor binding.

*For correspondence:
kyrgyzbala@gmail.com (SH);
ovcharen@nih.gov (IO)

**Competing interest:** The authors declare that no competing interests exist.

**Abstract** Enhancers and promoters are classically considered to be bound by a small set of transcription factors (TFs) in a sequence-specific manner. This assumption has come under increasing skepticism as the datasets of ChIP-seq assays of TFs have expanded. In particular, high-occupancy target (HOT) loci attract hundreds of TFs with often no detectable correlation between ChIP-seq peaks and DNA-binding motif presence. Here, we used a set of 1003 TF ChIP-seq datasets (HepG2, K562, H1) to analyze the patterns of ChIP-seq peak co-occurrence in combination with functional genomics datasets. We identified 43,891 HOT loci forming at the promoter (53%) and enhancer (47%) regions. HOT promoters regulate housekeeping genes, whereas HOT enhancers are involved in tissue-specific process regulation. HOT loci form the foundation of human super-enhancers and evolve under strong negative selection, with some of these loci being located in ultraconserved regions. Sequence-based classification analysis of HOT loci suggested that their formation is driven by the sequence features, and the density of mapped ChIP-seq peaks across TF-bound loci correlates with sequence features and the expression level of flanking genes. Based on the affinities to bind to promoters and enhancers we detected five distinct clusters of TFs that form the core of the HOT loci. We report an abundance of HOT loci in the human genome and a commitment of 51% of all TF ChIP-seq binding events to HOT locus formation thus challenging the classical model of enhancer activity and propose a model of HOT locus formation based on the existence of large transcriptional condensates.

## Introduction

Tissue -specificity of gene expression is orchestrated by the combination of transcription factors (TFs) that bind to regulatory regions such as promoters, enhancers, and silencers (*Moore et al., 2020*; *Gorkin et al., 2020*). Classically, an enhancer is thought to be bound by a few TFs that recognize a specific DNA motif at their cognate TF binding site (TFBS) through its DNA-binding domain and recruit other molecules necessary for catalyzing the transcriptional machinery (*Forsberg and Westin, 1991*; *Serfling et al., 1985*; *Sethi et al., 2020*). Based on the arrangements of the TFBSs, also called 'motif grammar', the architecture of enhancers is commonly categorized into 'enhanceosome' and

'billboard' models (*Spitz and Furlong, 2012*; *Long et al., 2016*). In the enhanceosome model, a rigid grammar of motifs facilitates the formation of a single structure comprising multiple TFs which then activates the target gene (*Thanos and Maniatis, 1995*; *Merika and Thanos, 2001*). This model requires the presence of all the participating proteins. Under the billboard model, on the other hand, the TFBSs are independent of each other and function in an additive manner (*Arnosti and Kulkarni, 2005*). However, as the catalogs of TF ChIP-seq assays have expanded thanks to the major collaborative projects such as ENCODE (*Davis et al., 2018*) and modENCODE (*Roy et al., 2010*), this assertion that the TFs interact with DNA through the strictly defined binding motifs has fallen under increasing contradiction with empirically observed patterns of DNA-binding regions of TFs. In particular, there have been reported genomic regions that seemingly get bound by a large number of TFs with no apparent DNA sequence specificity in terms of detectable binding motifs of corresponding motifs. These genomic loci have been dubbed high-occupancy target (HOT) regions and were detected in multiple species (*Roy et al., 2010*; *Moorman et al., 2006*; *Gerstein et al., 2010*; *Kvon et al., 2012*; *Yip et al., 2012*).

Initially, these regions have been partially attributed to technical and statistical artifacts of the ChIP-seq protocol, resulting in a small list of blacklisted regions that are mostly located in unstructured DNA regions such as repetitive elements and low complexity regions (*Teytelman et al., 2013*; *Wreczycka et al., 2019*). These blacklisted regions have been later excluded from the analyses and they represent a small fraction of the mapped ChIP-seq peaks. In addition, various studies have proposed the idea that some DNA elements can serve as permissive TF binding platforms such as GC-rich promoters, CpG islands, R-loops, and G-quadruplexes (*Teytelman et al., 2013*; *Wreczycka et al., 2019*). Other studies have concluded that these regions are highly functionally consequential regions enriched in epigenetic signals of active regulatory elements such as histone modification regions and high chromatin accessibility (*Roy et al., 2010*; *Ramaker et al., 2020*; *Partridge et al., 2020*).

Early studies of the subject have been limited in scope due to the small number of available TF ChIP-seq assays. There have been numerous studies in recent years with additional TFs across multiple cell lines. For instance, (*Partridge et al., 2020*), studied the HOT loci in the context of 208 proteins including TFs, cofactors, and chromatin regulators which they called chromatin-associated proteins. They observed that the composition of the chromatin-associated proteins differs depending on whether the HOT locus is located in an enhancer or promoter. *Wreczycka et al., 2019*, performed a cross-species analysis of HOT loci in the promoters of highly expressed genes, and established that some of the HOT loci correspond to the 'hyper-ChIPable' regions. *Ramaker et al., 2020*, conducted a comparative study of HOT regions in multiple cell lines and detected putative driver motifs at the core segments of the HOT loci.

In this study, we used the most up-to-date set of TF ChIP-seq assays available from the ENCODE Project (https://encodeproject.org/) and incorporated functional genomics datasets such as 3D chromatin data (Hi-C), eQTLs, GWAS, and clinical disease variants to characterize and analyze the functional implications of the HOT loci. We report that the HOT loci are one of the prevalent modes of regulatory TF-DNA interactions; they represent active regulatory regions with distinct patterns of bound TFs manifested as clusters of promoter-specific, enhancer-specific, and chromatin-associated proteins. They are active during the embryonic stage and are enriched in disease-associated variants. Finally, we propose a model for the HOT regions based on the idea of the existence of large transcriptional condensates.

## Results
### HOT loci are one of the prevalent modes of TF-DNA interactions

To define and analyze the HOT loci, we used the most up-to-date catalog of ChIP-seq datasets (n=1003) of TFs obtained from the ENCODE Project assayed in HepG2, K562, and H1-hESC (H1) cells (545, 411, and 47 ChIP-seq assays, respectively, see Methods for details). While the TFs are defined as sequence-specific DNA-binding proteins that control the transcription of genes, the currently available ChIP-seq datasets include the assays of many other types of transcription-related proteins such as cofactors, coactivators, histone acetyltransferases, as well as RNA Polymerase 2 variants. Therefore, we collectively call all of these proteins DNA-associated proteins (DAPs). Using the datasets of DAPs, we overlaid all of the ChIP-seq peaks and obtained the densities of DAP

binding sites across the human genome using a non-overlapping sliding window of length 400 bp and considered a binding site to be present in a given window if 8 bp centered at the summit of a ChIP-seq peak as overlapping. Given that the analyzed three cell lines contain varying numbers of assayed DAPs, we binned the loci according to the number of overlapping DAPs in a logarithmic scale with 10 intervals and defined HOT loci as those that fall to the highest four bins, which translates to those which contain on average >18% of available DAPs for a given cell line (see Methods for a detailed description and justifications). This resulted in 25,928, 15,231, and 2732 HOT loci in HepG2, K562, and H1 cells, respectively. We applied our definition to the Roadmap Epigenomic ChIP-seq datasets and observed that the number of available ChIP-seq datasets significantly affects the resulting HOT loci. However, the HOT loci defined using the Roadmap Epigenomic datasets were almost entirely composed of subsets of the ENCODE-based HOT loci, comprising 50%, 62%, and 15% in HepG2, K562, and H1, respectively (*Supplementary file 1, table S5*). Importantly, we note that the distribution of the number of loci is not multimodal, but rather follows a uniform spectrum, and thus, this definition of HOT loci is ad hoc (*Figure 1A*, *Figure 1—figure supplement 1*). Therefore, in addition to the dichotomous classification of HOT and non-HOT loci, we use all of the DAP-bound loci to extract the correlations with studied metrics with the number of bound DAPs when necessary. Throughout the study, we used the loci from the HepG2 cell line as the primary dataset for analyses and used the K562 and H1 datasets when the comparative analysis was necessary.

Although the HOT loci represent only 5% of all the DAP-bound loci in HepG2, they contain 51% of all mapped ChIP-seq peaks. The fraction of the ChIP-seq peaks of each DAP overlapping HOT loci varies from 0% to 91%, with an average of 65% (*Figure 1B*, y-axis). Among the DAPs that are present in the highest fraction of HOT loci are (*Figure 1B*, x-axis) SAP130, MAX, ARID4B, ZGPAT, HDAC1, MED1, TFAP4, and SOX6. The abundance of histone deacetylase-related factors mixed with transcriptional activators suggests that the regulatory functions of HOT loci are a complex interplay of activation and repression. RNA Polymerase 2 (POLR2) is present in 42% of HOT loci arguing for active transcription at or in the proximity of HOT loci (including mRNA and eRNA transcription). When the fraction of peaks of individual DAPs overlapping with the HOT loci are considered (*Figure 1B*, y-axis), DAPs with >90% overlap are GMEB2 (essential for replication of parvoviruses), ZHX3 (zinc finger transcriptional repressor), and YEATS2 (subunit of acetyltransferase complex). Whereas the DAPs that are least associated with HOT loci (<5%) are ZNF282 (transcriptional repressor), MAFK, EZH2 (histone methyltransferase), and TRIM22 (ubiquitin ligase). The fact that HOT loci harbor more than half of the ChIP-seq peaks suggests that the HOT loci are one of the prevalent modes of TF-DNA interactions rather than an exceptional case, as has been initially suggested by earlier studies (*Teytelman et al., 2013*; *Wreczycka et al., 2019*).

Around half of the HOT loci (51%) are located in promoter regions (46% in primary promoters and 5% in alternative promoters), 25% in intronic regions, and only 24% are in intergenic regions with 9% being located >50 kb away from promoters, suggesting that the HOT loci are mainly clustered in vicinities (promoters and introns) of transcription start sites and therefore potentially playing essential roles in the regulation of nearby genes (*Figure 1C*). When considering the non-promoter HOT loci, we observed that they were universally located in regions of H3K27ac or H3K4me1, indicating that they are active enhancers (*Figure 1—figure supplement 2*). When comparing the definitions of promoters and enhancers based on chromHMM states and ENCODE SCREEN annotations, the composition of HOT loci in relation to promoters and enhancers showed similar fractions (*Figure 1—figure supplement 3*). Both HOT promoters and enhancers are almost entirely located in the chromatin-accessible regions (97% and 93% of the total sequence lengths, respectively, *Figure 1D*). We compared our definition of the HOT loci to those reported in *Ramaker et al., 2020*, and *Boyle et al., 2014*. We observed that because these two studies define HOT loci using 2 kb windows, they cover a larger fraction of the genome. Our set of HOT loci largely consisted of subsets of those defined in these two studies, with overlap percentages of 81%, 93%, and 100% in HepG2, K562, and H1, respectively (*Figure 1—figure supplement 4*). Further analysis revealed that our set of HOT loci primarily constitutes the 'core' and more conserved (*Figure 1—figure supplement 5*) regions of HOT loci defined in the mentioned studies, while their composition in terms of promoter, intronic, and intergenic regions is similar (*Figure 1—figure supplement 6*), suggesting that the three definitions point to loci with similar characteristics.

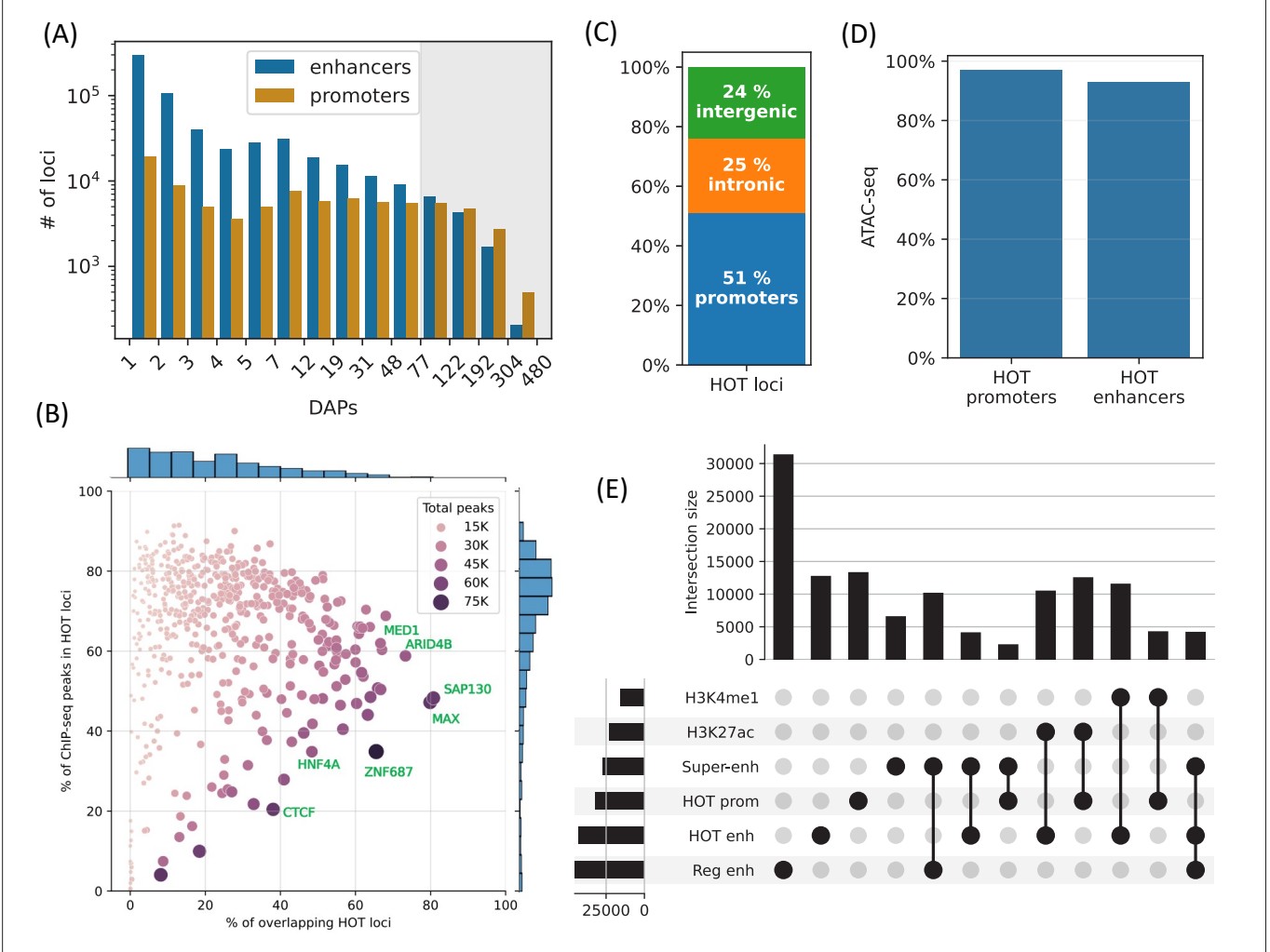

**Figure 1.** High-occupancy target (HOT) loci are prevalent in the genome. (**A**) Distribution of the number of loci by the number of overlapping peaks 400 bp loci. Loci are binned on a logarithmic scale (*Table 1*, Methods). The shaded region represents the HOT loci. (**B**) Prevalence of DNA-associated proteins (DAPs) in HOT loci. Each dot represents a DAP. X-axis: percentage of HOT loci in which DAP is present (e.g. MAX is present in 80% of HOT loci). Y-axis: percentage of total peaks of DAPs that are located in HOT loci (e.g. 45% of all the ChIP-seq peaks of MAX is located in the HOT loci). Dot color and size are proportional to the total number of ChIP-seq peaks of DAP. (**C**) Breakdown of HepG2 HOT loci to the promoter, intronic, and intergenic regions. (**D**) Fractions of HOT enhancer and promoter loci located in ATAC-seq. (**E**) Overlaps between the HOT enhancer, HOT promoter, super-enhancer, regular enhancer, H3K27ac, and H4K4me1 regions. Horizontal bars on bottom left represent the total number of loci of the corresponding class of loci. All of the visualized data is generated from the HepG2 cell line.

The online version of this article includes the following figure supplement(s) for figure 1:

**Figure supplement 1.** Distribution of the number of loci by the number of overlapping peaks 400 bp loci in K562 and H1.

**Figure supplement 2.** Percentages of overlapping promoter (top row) and enhancer (bottom row) loci binned by bound DNA-associated proteins (DAPs) with histone modification regions in HepG2 (left column) and K562 (right column).

**Figure supplement 3.** Composition of the high-occupancy target (HOT) loci to promoter and enhancer regions based on the definitions used in this study, chromHMM states and ENCODE SCREEN annotations.

**Figure supplement 4.** Overlaps between the high-occupancy target (HOT) loci as reported in this study (*Ramaker et al., 2020* and *Boyle et al., 2014*).

**Figure supplement 5.** phastCons conservation scores of high-occupancy target (HOT) loci defined by this study (*Ramaker et al., 2020* and *Boyle et al., 2014*).

**Figure supplement 6.** Compositions of high-occupancy target (HOT) loci as reported in this study (*Ramaker et al., 2020* and *Boyle et al., 2014*) in terms of promoter, intronic, and intergenic regions.

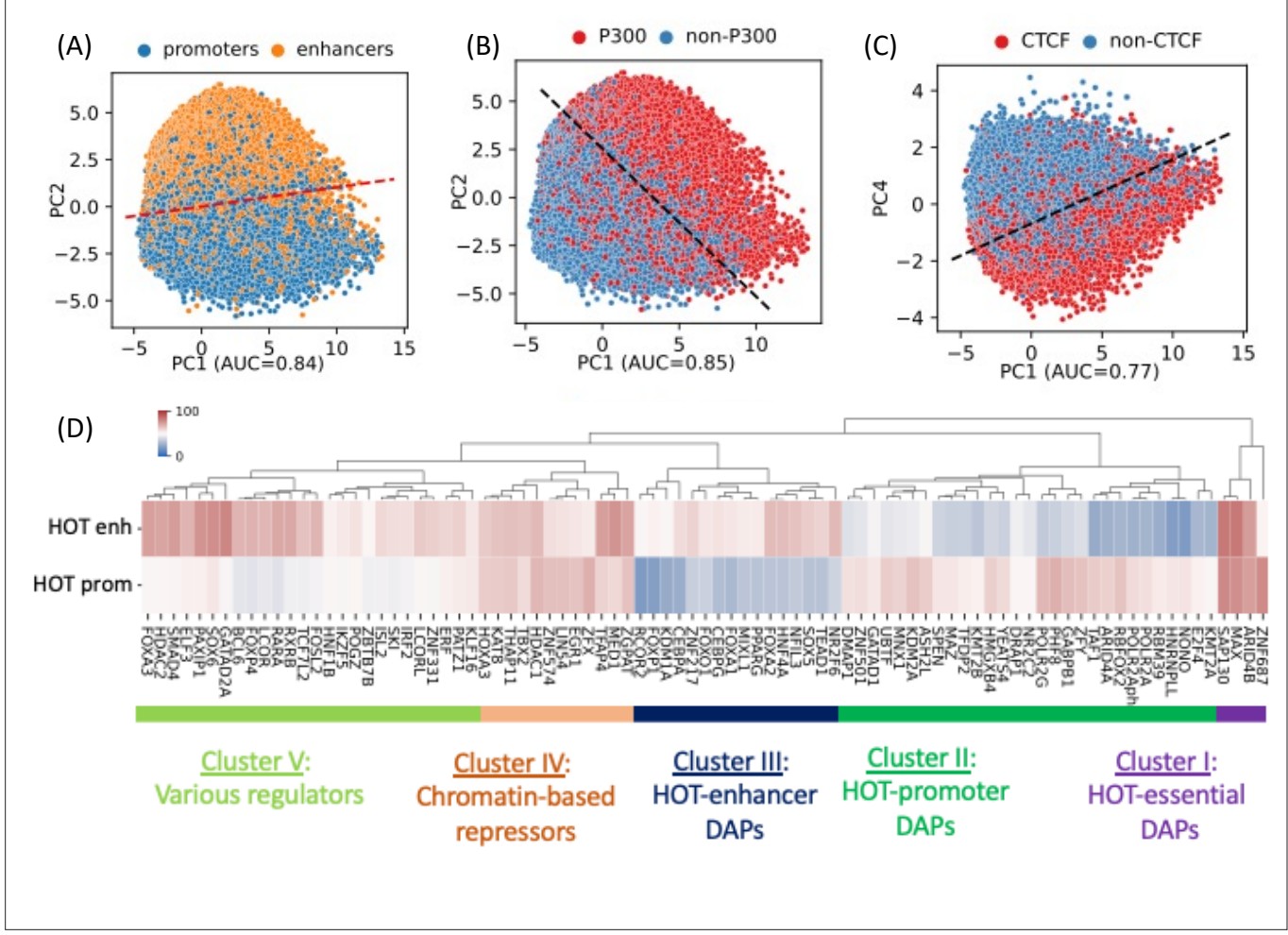

**Figure 2.** PCA plots of high-occupancy target (HOT) loci based on the DNA-associated protein (DAP) presence vectors. Each dot represents a HOT locus: (**A**) PC1 and PC2, marked promoters and enhancers. (**B**) PC1 and PC2, marked p300-bound HOT loci. (**C**) PC1 and PC4, marked CTCF-bound HOT loci. The dashed lines in A, B, C are logistic regression lines. auROC values are results of logistic regression. (**D**) DAPs hierarchically clustered by their involvement in HOT promoters and HOT enhancers. Heatmap colors indicate the % of HOT enhancers or promoters that a given DAP overlaps with. All of the visualized data is generated from the HepG2 cell line.

The online version of this article includes the following figure supplement(s) for figure 2:

**Figure supplement 1.** PCA plots of high-occupancy target (HOT) loci in HepG2 based on the DNA-associated protein (DAP) presence vectors.

**Figure supplement 2.** DNA-associated proteins (DAPs) clustered by percentage of high-occupancy target (HOT) promoters and HOT enhancers that the ChIP-seq peaks overlap.

**Figure supplement 3.** PPI networks of four clusters.

**Figure supplement 4.** CTCF and Cohesin in high-occupancy target (HOT) loci.

To further dissect the composition of HOT enhancer loci, we compared them to super-enhancers as defined in the study by *Whyte et al., 2013*, and a set of regular enhancers (Methods). Overall, 31% of HOT enhancers and 16% of HOT promoters are located in super-enhancers, while 97% of all HOT loci overlap with H3K27ac or H3K4me1 regions (*Figure 1E*). While HOT enhancers and promoters appear to provide a critical foundation for super-enhancer formation, they represent only a small fraction of super-enhancer sequences overall accounting for 9% of combined super-enhancer length.

A 400 bp HOT locus, on average, harbors 125 DAP peaks in HepG2. However, the peaks of DAPs are not uniformly distributed across HOT loci. There are 68 DAPs with >80% of all of the peaks located in HOT loci (*Figure 1B*). To analyze the signatures of unique DAPs in HOT loci, we performed a PCA where each HOT locus is represented by a binary (presence/absence) vector of length equal to the total number of DAPs analyzed. This analysis showed that the principal component 1 (PC1) is

correlated with the total number of distinct DAPs located at a given HOT locus (*Figure 2—figure supplement 1A*). PC2 separates the HOT promoters and HOT enhancers (*Figure 2A*, *Figure 2—figure supplement 1B*), and the PC1-PC2 combination also separates the p300-bound HOT loci (*Figure 2B*, *Figure 2—figure supplement 1C*). This indicates that the HOT promoters and HOT enhancers must have distinct signatures of DAPs. To test if such signatures exist, we clustered the DAPs according to the fractions of HOT promoter and HOT enhancer loci that they overlap with. This analysis showed that there is a large cluster of DAPs (n=458) which on average overlap with only 17% of HOT loci which are likely secondary to the HOT locus formation (*Figure 2—figure supplement 2*). We focused on the other, HOT-enriched, cluster of DAPs (n=87) which are present in 53% of HOT loci on average (*Figure 2—figure supplement 2*) and consist of four major clusters of DAPs (*Figure 2D*). *Cluster I* comprises four DAPs ZNF687, ARID4B, MAX, and SAP130 which are present in 75% of HOT loci on average. The three latter of these DAPs form a PPI interaction network (PPI enrichment p-value=0.001) (*Figure 2—figure supplement 3A*). We called this cluster of DAPs essential regulators given their widespread presence in both HOT enhancers and HOT promoters. *Cluster II* comprises 29 DAPs which are present in 47% of the HOT loci and are 1.7× more likely to overlap with HOT promoters than HOT enhancers. Among these DAPs are POLR2 subunits, PHF8, GABP1, GATAD1, TAF1, etc. The strongest associated GO molecular function term with the DAPs of this cluster is *RNA Polymerase transcription factor initiation activity* suggestive of their direct role in transcriptional activity (*Figure 2—figure supplement 3B*). *Cluster III* comprises 16 DAPs which are 1.9× more likely to be present in HOT enhancers than in HOT promoters. These are a wide variety of transcriptional regulators among which are those with high expression levels in liver NFIL3, NR2F6, and pioneer factors HNF4A, CEBPA, FOXA1, and FOXA2. The majority (13/16) of DAPs of this cluster form a PPI network (PPI enrichment p-value<$10^{-16}$, *Figure 2—figure supplement 3C*). Among the strongest associated GO terms of biological processes are those related to cell differentiation (*white fat cell differentiation*, *endocrine pancreas development*, *dopaminergic neuron differentiation*, etc.) suggesting that *cluster III* HOT enhancers underlie cellular development. *Cluster IV* comprises 12 DAPs which are equally abundant in both HOT enhancers and HOT promoters (64% and 63%, respectively), which form a PPI network (PPI enrichment p-value<$10^{-16}$, *Figure 2—figure supplement 3D*) with HDAC1 (histone deacetylase 1) being the node with the highest degree, suggesting that the DAPs of the cluster may be involved in chromatin-based transcriptional repression. Lastly, *Cluster V* comprises 26 DAPs of a wide range of transcriptional regulators, with a 1.3× skew toward the HOT enhancers. While this cluster contains prominent TFs such as TCF7L2, FOXA3, SOX6, FOSL2, etc., the variety of the pathways and interactions they partake in makes it difficult to ascertain the functional patterns from the constituent of DAPs alone. Although this clustering analysis reveals subsets of DAPs that are specific to either HOT enhancers or HOT promoters (Clusters II and III), it still does not explain what sorts of interplays take place between these recipes of HOT promoters and HOT enhancers, as well as with the other clusters of DAPs with equal abundance in both the HOT promoters and HOT enhancers.

Notably, PC4 separates HOT loci associated with CTCF (*Figure 2C*) and Cohesin (*Figure 2—figure supplement 1D*). This clear separation of CTCF- and Cohesin-bound HOTs is surprising, given that only relatively small fractions of their peaks (21% and 38%, respectively) reside in HOT loci, and present in 36% of the HOT loci, compared to some other DAPs with much higher presence described above, that do not get separated clearly by the PCA. Furthermore, CTCF- and Cohesin-bound HOT enhancer loci are located significantly closer (p-value<$10^{-100}$; Mann-Whitney U test) to the nearest genes (*Figure 2—figure supplement 4A*), making it more likely that those loci are proximal enhancers. And the total number of overlapping DAPs is significantly higher (p-value<$10^{-100}$; Mann-Whitney U test) in CTCF- and Cohesin-bound loci compared to the rest of the HOT loci (*Figure 2—figure supplement 4B*), suggesting that at least a portion of the number of DAPs in HOT loci can be explained by 3D chromatin contacts between the genomic regions mediated by CTCF-Cohesin complex.

To comprehensively quantify the 3D chromatin interactions involving the HOT loci, we used Hi-C data with 5 kb resolution (*Lieberman-Aiden et al., 2009*) (see Methods). First, we obtained statistically significant chromatin interactions using FitHiChIP tool (*Bhattacharyya et al., 2019*) (see Methods) and observed that HOT loci are enriched in chromatin interactions and 1.66× more likely to engage in chromatin interactions than the regular enhancers (p-value<$10^{-20}$, Chi-square test). When all of the DAP-bound loci are considered, the number of chromatin interactions positively correlates with the number of bound DAPs (rho = 0.3, p-value<$10^{-100}$, Spearman correlation). Next, we overlayed the

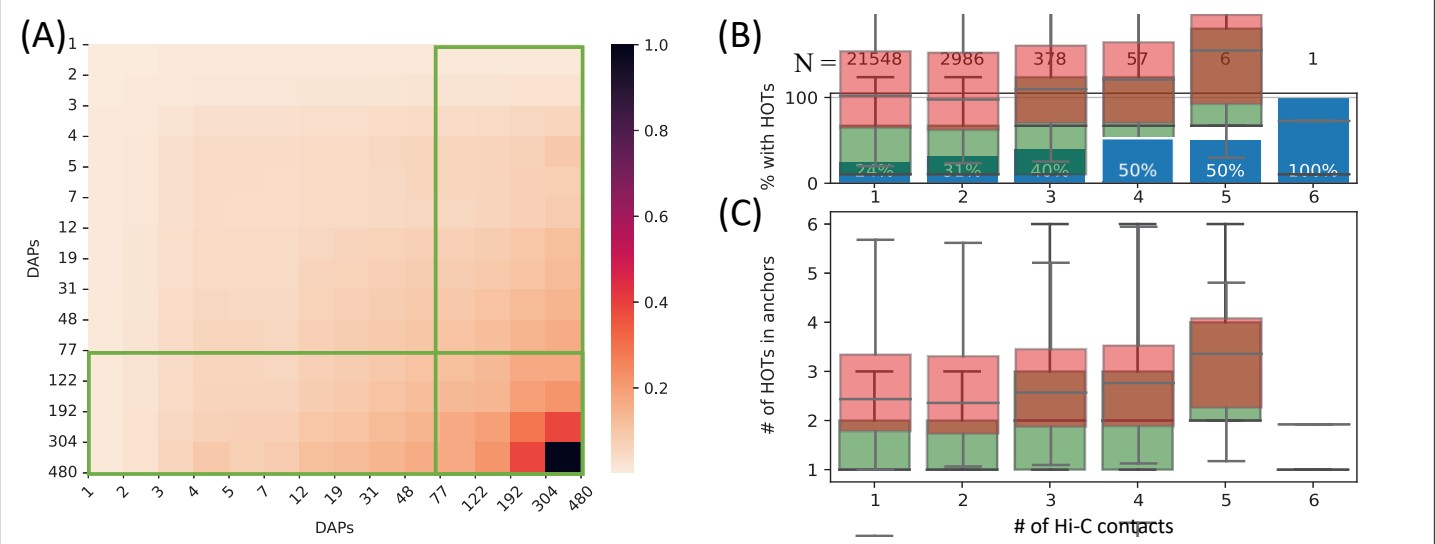

**Figure 3.** High-occupancy target (HOT) loci in high-frequency 3D chromatin interaction regions. (**A**) Densities of long-range Hi-C chromatin contacts between the DNA-associated protein (DAP)-bound loci. Each horizontal and vertical bin represents the loci with the number of bound DAPs between the edge values. The density values of each cell are normalized by the maximum value across all pairwise bins. Green boxes represent HOT loci. (**B**) Distribution of HOT loci in Hi-C contact regions. X-axis is the number of Hi-C contacts. Numbers in the top row indicate the total number of genomic loci engaging in the given number of Hi-C contacts. Bars indicate the % of Hi-C loci that contain at least one HOT locus. (**C**) Distribution of the number of HOT loci in regions with a given number of Hi-C contacts. X-axis is the same as B. All of the visualized data is generated from the HepG2 cell line.

chromatin interactions with the loci binned by the number of bound DAPs. We observed that the loci with high numbers of bound DAPs are more likely to engage in chromatin interactions with other loci harboring large numbers of DAPs, i.e., the HOT loci have the propensity to connect through long-range chromatin interactions with other HOT loci (*Figure 3A*). To further validate this observation, we obtained frequently interacting regions (FIREs) (*Schmitt et al., 2016*), and observed that the FIREs are 2.89× (p-value<10$^{-230}$, Chi-square test) enriched HOT loci compared to the regular enhancers (see Methods). Moreover, 66% of HOT loci are located in TAD regions and 21% are located in chromatin loops. In particular, the HOT loci are 2.97× (p-value<10$^{-230}$, Mann-Whitney U test) enriched in the chromatin loop anchor regions (11% of the HOT loci) compared to regular enhancers. To investigate further, we analyzed the loop anchor regions harboring HOT loci and observed that the number of multi-way contacts on loop anchors (i.e. loci that serve as anchors to multiple loops) correlates with the number of bound DAPs (rho = 0.84 p-value<10$^{-4}$; Pearson correlation). The number of multi-way interactions in loop anchor regions varies between 1 and 6, with only one locus, in an extreme case, serving as an anchor for 6 overlapping loops on chromosome 2 (*Figure 3B*). Of the loop anchor regions with >3 overlapping loops, more than half contained at least one HOT locus, suggesting an interplay between chromatin loops and HOT loci (*Figure 3B*). Overall, 94% of HOT loci are located in regions with at least one chromatin interaction. This observation is consistent with previous reports that much of the long-range 3D chromatin contacts form through the interactions of large protein complexes (*Quinodoz et al., 2018*). While there is a correlation between the HOT loci and chromatin interactions, the causal relation between these two properties of genomic loci is not clear.

## A set of DAPs stabilizes the interactions of DAPs at HOT loci

Next, we sought to analyze the patterns of ChIP-seq signal values at HOT loci, as a metric for overall DAP occupancy at genomic loci. We observed that the overall signals of DAPs correlate with the total number of colocalizing DAPs (*Figure 4A*, rho = 0.97, p-value<10$^{-10}$; Spearman correlation). Moreover, even when calculated DAP-wise, the average of the overall signal strength of every DAP correlates with the fraction of HOT loci that the given DAP overlaps with (rho = 0.6, p-value<10$^{-29}$; Spearman correlation, *Figure 4B*), meaning that the overall average value of the signal intensity of a given DAP is largely driven by the ChIP-seq peaks which are located in HOT loci.

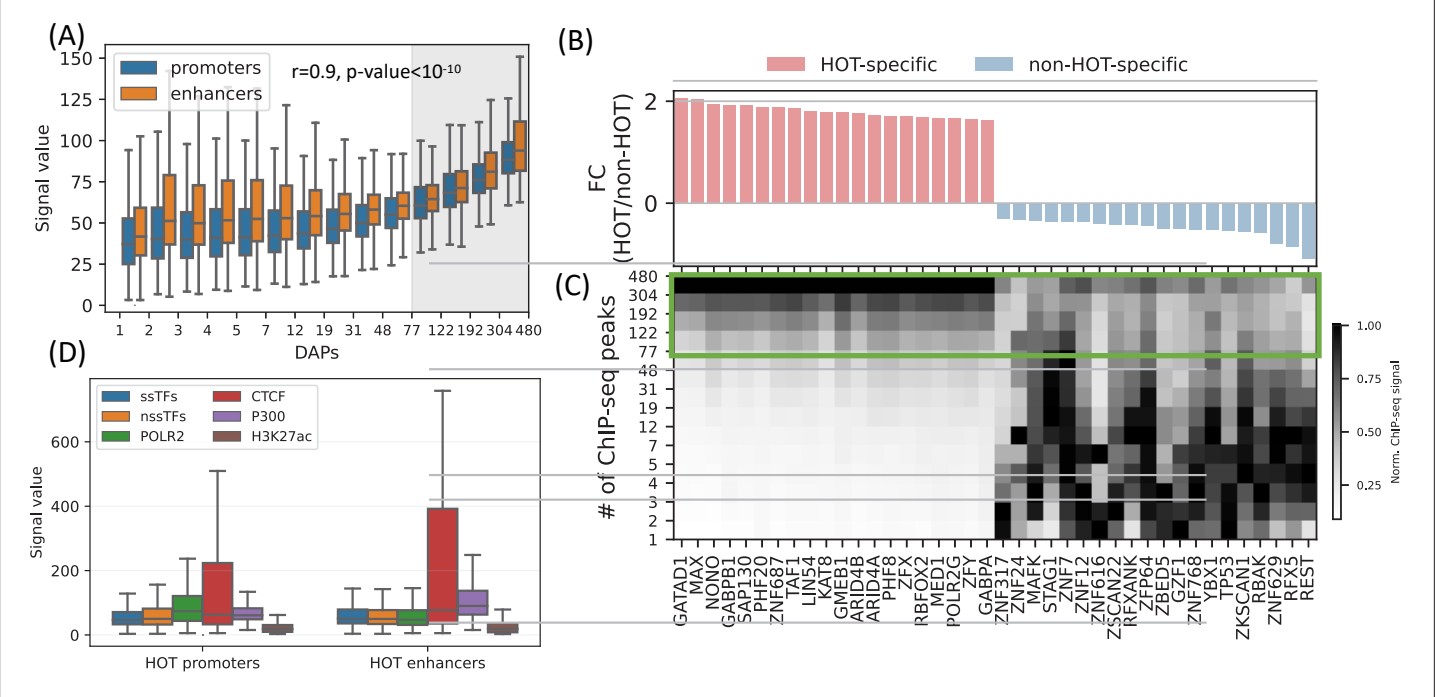

**Figure 4.** High-occupancy target (HOT) regions induce strong ChIP-seq signals. (**A**) Distribution of the signal values of the ChIP-seq peaks by the number of bound DNA-associated proteins (DAPs). The shaded region represents the HOT loci. (**B, C**) DAPs sorted by the ratio of ChIP-seq signal strength of the peaks located in HOT loci and non-HOT loci. 20 most HOT-specific (red bars) and 20 most non-HOT-specific (blue bars) DAPs are depicted. (**B**) Fold-change (log2) of the HOT and non-HOT loci ChIP-seq signals. (**C**) Distribution of the average ChIP-seq signal in the loci binned by the number of bound DAPs. Rows represent the loci with the bound DAPs indicated by the values of the edges (y-axis). Green box regions demarcate the HOT regions. (**D**) Signal values of sequence-specific DAPs (ssDAPs), non-sequence-specific DAPs (nssDAPs) (see the text for description), H3K27ac, CTCF, P300 peaks in HOT promoters and enhancers. All of the visualized data is generated from the HepG2 cell line.

The online version of this article includes the following figure supplement(s) for figure 4:

**Figure supplement 1.** Normalized ChIP-seq signal values of DNA-associated proteins (DAPs) in high-occupancy target (HOT) loci (rows) in the presence of other DAPs (columns).

**Figure supplement 2.** Distribution of the ChIP-seq signal values of DNA-associated proteins (DAPs) when the stabilizing DAPs (*Figure 4—figure supplement 1*) are present vs. absent.

While the overall average of the ChIP-seq signal intensity in HOT loci is greater when compared to the rest of the DAP-bound loci, individual DAPs demonstrate different levels of involvement in HOT loci. When sorted by the ratio of the signal intensities in HOT vs. non-HOT loci, among those with the highest HOT-affinities are GATAD1, MAX, NONO, as well as POLR2G and Mediator subunit MED1 (*Figure 4B and C*). Whereas those with the opposite affinity (i.e. those that have the strongest binding sites in non-HOT loci) are REST, RFX5, TP53, etc. (*Figure 4B and C*). By analyzing the signal strengths of DAPs jointly, we observed that a host of DAPs likely has a stabilizing effect on the binding of DAPs in that, when present, the signal strengths of the majority of DAPs are on average 1.9× greater (p-value<$10^{-100}$, Mann-Whitney U test). These DAPs are CREB1, RFX1, ZNF687, RAD51, ZBTB40, and GPBP1L1 (Appendix 1 – Joint DAPs analysis, *Figure 4—figure supplements 1 and 2*).

So far, we have treated the DAPs under a single category and did not make a distinction based on their known DNA-binding properties. Previous studies have discussed the idea that sequence-specific DAPs (ssDAPs) can serve as anchors, similar to the pioneer TFs, which could facilitate the formation of HOT loci (*Ramaker et al., 2020*; *Partridge et al., 2020*; *Xie et al., 2013*). We asked if ssDAPs yield greater signal strength values than non-sequence-specific DAPs (nssDAPs). To test this hypothesis, we classified the DAPs into those two categories using the definitions provided in the study (*Lambert et al., 2018*), where the TFs are classified by curation through extensive literature review and supported by annotations such as the presence of DNA-binding domains and validated binding motifs. Based on this classification, we categorized the ChIP-seq signal values into these two groups. While statistically significant (p-value<0.001, Mann-Whitney U test), the differences in the average

signals of ssDAPs and nssDAPs in both HOT enhancers and HOT promoters are small (*Figure 4D*). Moreover, while the average signal values of ssDAPs in HOT enhancers are greater than that of the nssDAPs, in HOT promoters this relation is reversed. At the same time, the average signal strength of the DAPs is 3× greater than the average signal strength of H3K27ac peaks in HOT loci. Based on this, we concluded that the ChIP-seq signal intensities do not seem to be a function of the DNA-binding properties of the DAPs.

## Sequence features that drive the accumulation of DAPs

We next analyzed the sequence features of the HOT loci. For this purpose, we first addressed the evolutionary conservation of the HOT loci using phastCons scores generated using an alignment of 46 vertebrate species (*Siepel et al., 2005*). The average conservation scores of the DAP-bound loci are in strong correlation with the number of bound DAPs (rho = 0.98, p-value<$10^{-130}$; Spearman correlation), indicating that the negative selection exerted on HOT loci are proportional to the number of bound DAPs (*Figure 5A*). With 120 DAPs per locus on average, these HOT regions are 1.7× more conserved than the regular enhancers in HepG2 (*Figure 5B*). We observed a similar trend of conservation levels when the phastCons scores generated from primates and placental mammals and primates were considered, the HOT loci being 1.45× and 1.1× more conserved than the regular enhancers, respectively (*Figure 5—figure supplement 1*). In addition, we observed that the HOT loci of all three cell lines (HepG2, K562, and H1) overlap with 22 ultraconserved regions, among which are the promoter regions of 11 genes including SP5, SOX5, AUTS2, PBX1, ZFPM2, ARID1A, OLA1 and the enhancer regions of (within <50 kb of their TSS) 5S rRNA, MIR563, SOX21, etc. (full list in *Supplementary file 1, table S4*). Among them are those which have been linked to diseases and other phenotypes. For example, DNAJC1 (*Michailidou et al., 2017*) and OLA1 (which interacts with BRCA1) have been linked to breast cancer in cancer GWAS studies (*Liu et al., 2020*). Whereas AUTS2 (*Biel et al., 2022*) and SOX5 (*Schanze et al., 2013*) have been linked to predisposition to neurological conditions such as autism spectrum disorder, intellectual disability, and neurodevelopmental disorder. Of these genes, ARID1A, AUTS2, DNAJC1, OLA1, SOX5, and ZFPM2 have been reported to have strong activities in the Allen Mouse Brain Atlas (*Daigle et al., 2018*).

CpG islands have been postulated to serve as permissive TF binding platforms (*Pachano et al., 2021*; *Deaton and Bird, 2011*) and this has been listed as one of the possible reasons for the existence of HOT loci in a previous study (*Wreczycka et al., 2019*). To test this hypothesis, we extracted the overlap rates of all DAP-bound loci with CpG islands (Methods). While the overall fraction of loci that overlap CpG islands correlates strongly with the number of bound DAPs (rho = 0.7, p-value=0.001; Pearson correlation), only 12% of HOT enhancers overlapped CpG island whereas, for the HOT promoters, this fraction was 83%, suggesting that CpG islands alone do not explain HOT enhancer loci despite accounting for the majority of HOT promoters loci (*Figure 5—figure supplement 2A*). Similarly, the average GC content is strongly correlated with the number of bound DAPs (rho = 0.89, p-value<$10^{-4}$; Pearson correlation, *Figure 5—figure supplement 2B*), with the average GC content of 64% and 51% in HOT promoters and HOT enhancers respectively (p-value<$10^{-100}$, Mann-Whitney U test), in both HepG2 and K562.

In addition, we observed that the average content of repeat elements in the loci strongly and negatively correlates with the number of bound DAPs across the cell lines (rho = −0.9, p-value=<$10^{-5}$; Pearson, *Figure 5—figure supplement 2C*), which is likely the result of the fact that the HOTs are under elevated negative selection and reject insertion of repetitive DNA.

Other genomic sequence features that have been considered in the context of HOT loci in previous studies include and are not limited to G-quadruplex, R-loops, methylation patterns, etc., which have concluded that each of them can partially explain the phenomenon of the HOT loci (*Moorman et al., 2006*; *Teytelman et al., 2013*; *Wreczycka et al., 2019*). Still, one of the central questions remains whether the HOT loci are driven by sequence features or they are the result of cellular biology not strictly related to the sequences, such as the proximal accumulation of DAPs in foci due to the biochemical properties of accumulated molecules, or other epigenetic mechanisms.

To address this question with a broader approach, we asked whether the HOT loci can be accurately predicted based on their DNA sequences alone, and sequence features, including GC, CpG, GpC contents, and CpG island coverage. For sequence-based classification, we trained a convolutional neural network (CNN) model using one-hot encoded sequences and an SVM classifier trained

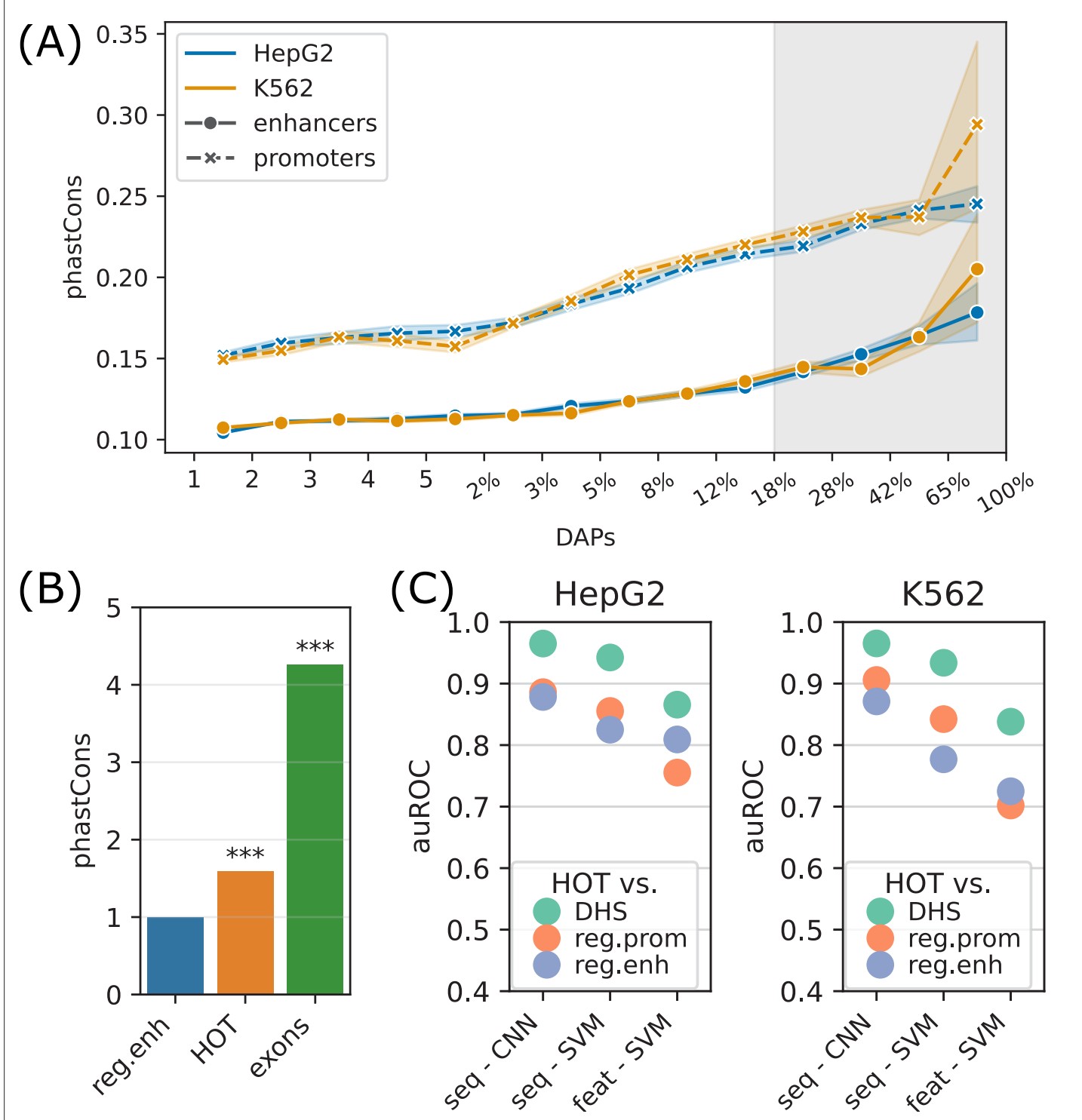

**Figure 5.** Sequence features of high-occupancy target (HOT) loci. (**A**) Distribution of conservation score in loci bound by DNA-associated proteins (DAPs) in HepG2 and K562. The logarithmic part of the bins is expressed in terms of the percentages of loci that each bin covers, averaged over two cell lines. The shaded region represents HOT loci. (**B**) phastCons conservation scores of regular enhancer, HOT loci, and exon regions. The values are normalized by the average scores of regular enhancers. (**C**) Classification performances (auROC) of HOT loci against the backgrounds of DNase-I hypersensitivity sites (DHS), promoter, and regular enhancer regions. The x-axis values are the methods used for classifications. Methods starting with 'seq -' are based on sequences (convolutional neural networks [CNNs] and gkmSVM). Starting with 'feat -' are methods where all sequence features are used (GC, CpG, GpC, CpG island).

*Figure 5 continued on next page*

Hudaiberdiev and Ovcharenko. eLife 2024;13:RP95170. DOI: https://doi.org/10.7554/eLife.95170                    10 of 30

*Figure 5 continued*

The online version of this article includes the following figure supplement(s) for figure 5:

**Figure supplement 1.** Comparison of phastCons conservation scores of regular enhancers, high-occupancy target (HOT) loci, and exons using the score extracted from vertebrates, placental mammals, and primates.

**Figure supplement 2.** Sequence features of high-occupancy target (HOT) loci.

**Figure supplement 3.** Comparison of classification performances for sequences in different lengths.

**Figure supplement 4.** Distances and expressions of flanking genes of DNA-associated protein (DAP)-bound loci.

on gapped k-mers (seq-SVM) (*Lee, 2016*). Using the sequence features we trained SVM models with linear kernel function (feature-SVM). We carried out the classification experiments using the following control (i.e. negative) sets: (a) randomly selected loci from merged DNase I hypersensitivity sites (DHS) of cell lines in the Roadmap Epigenomics Project, (b) promoter regions, and (c) regular enhancers. When averaged over cell lines and control sets, CNN, seq-SVM, and feature-SVM models yielded auROC values of 0.91, 0.86, and 0.78 respectively, suggesting that CNNs capture the motif grammar of the HOT loci better than the compared models (*Figure 5C*). The superiority of sequence-based models over feature-based classification by a factor of 1.3× (or 17%) suggests that there is additional information that is highly relevant to the DNA-DAP interaction density encoded in the DNA sequences, in addition to the GC, CpG, GpC contents. (See Appendix 1 – Classification results analyses for further details of model training, and comparison of performances of different combinations of SVM kernels and feature sets, as well as Logistic Regression as a baseline.) This is in line with the observation mentioned above, that 88% of the HOT enhancers do not overlap with annotated CpG islands. This analysis concluded that the mechanisms of HOT locus formation are likely encoded in their DNA sequences.

Extending the input regions from 400 bp to 1 kb for sequence-based classification did not lead to a significant increase in performance, suggesting that the core 400 bp regions contain most of the information associated with DAP density (*Figure 5—figure supplement 3*).

## Highly expressed housekeeping genes are commonly regulated by HOT promoters

After characterizing the HOT loci in terms of the DAP composition and sequence features, we sought to analyze the cellular processes they partake in. HOT loci were previously linked to highly expressed genes (*Wreczycka et al., 2019*). In both inspected differentiated cell lines (HepG2 and K562), the number of DAPs positively correlates with the expression level of their target gene (enhancers were assigned to their nearest genes for this analysis; rho = 0.56, p-value<$10^{-10}$; Spearman correlation; *Figure 5—figure supplement 4A*). In HepG2, the average expression level of the target genes of promoters with at least one DAP bound is 1.7× higher than that of the target genes of enhancers with at least one DAP bound, whereas when only HOT loci are considered this fold-increase becomes 4.7×. This suggests that the number of bound DAPs of the HOT locus has a direct impact on the level of the target gene expression. Moreover, highly expressed genes (RPKM>50) were 4× more likely to have multiple HOT loci within the 50 kb of their TSSs than the genes with RPKM<5 (p-value<$10^{-12}$, Chi-square test). In addition, the average distance between HOT enhancer loci and the nearest gene is 4.5× smaller than with the regular enhancers (p-value<$10^{-30}$, Mann-Whitney U test). Generally, we observed that the distances between the HOT enhancers and the nearest genes are negatively correlated with the number of bound DAPs (rho = −0.9; p-value<$10^{-6}$; Pearson correlation; *Figure 5—figure supplement 4B*), suggesting that the increasing number of bound DAPs makes the regulatory region more likely to be the TSS-proximal regulatory region.

To further analyze the distinction in involved biological functions between the HOT promoters and enhancers, we compared the fraction of housekeeping (HK) genes that they regulate, using the list of HK genes reported by *Hounkpe et al., 2021*. According to this definition, 64% of HK genes are regulated by a HOT promoter and only 30% are regulated by regular promoters (*Figure 6A*). The HOT enhancers, on the other hand, flank 21% of the HK genes, which is less than the percentage of HK genes flanked by regular enhancers (38%). For comparison, 22% of the flanking genes of super-enhancers constitute HK genes. The involvement of HOT promoters in the regulation of HK genes is also confirmed in terms of the fraction of loci flanking the HK genes, namely, 21% of the HOT

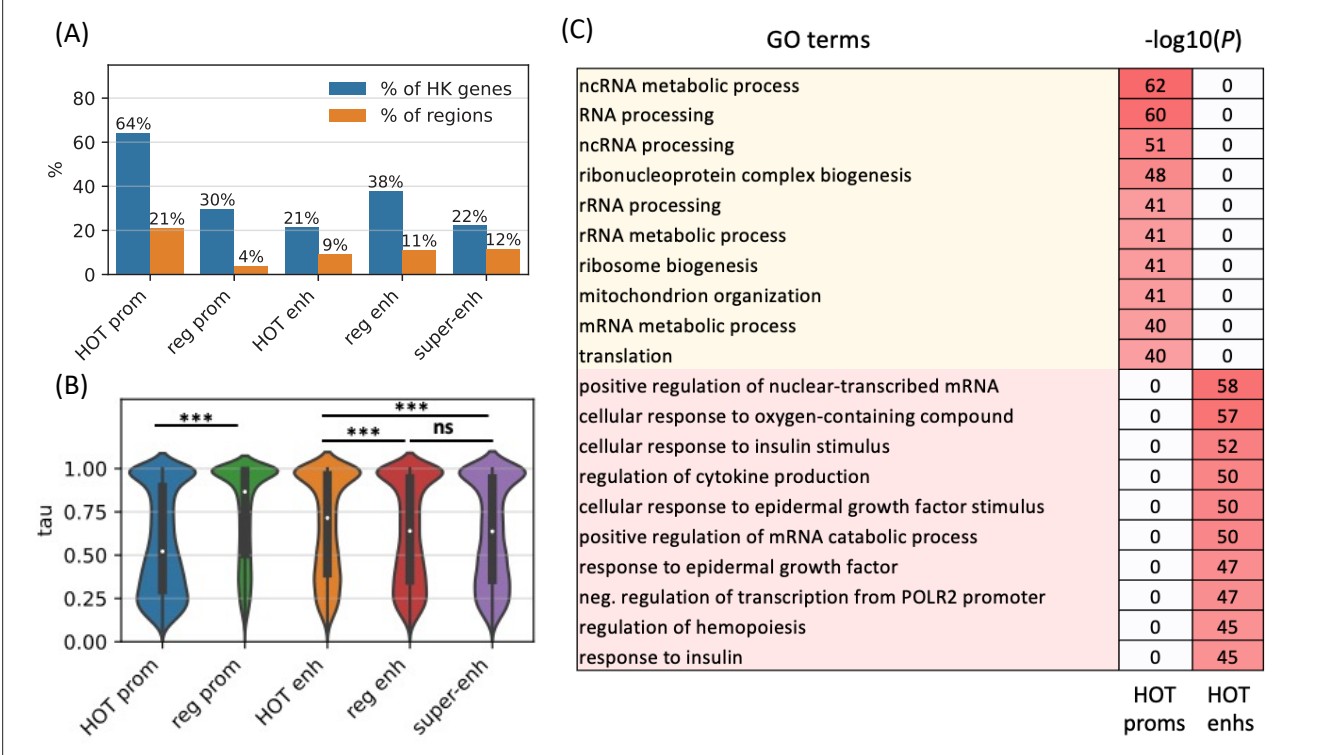

**Figure 6.** High-occupancy target (HOT) promoters are ubiquitous and HOT enhancers are tissue-specific. (**A**) Fractions of housekeeping genes regulated by the given category of loci (blue). Fractions of the loci which regulate the housekeeping genes (orange). (**B**) Tissue specificity (*tau*) scores of the target genes of different types of regulatory regions. (**C**) GO enriched terms of HOT promoters and enhancers of HepG2. 0 values in the p-values columns indicate that the GO term was not present in the top 50 enriched terms as reported by the GREAT tool. All of the visualized data is generated from the HepG2 cell line.

promoters regulate 64% of the HK genes. This fraction is much smaller (<9% on average) for the rest of the mentioned categories of loci (HOT and regular enhancers, regular promoters, and super-enhancers, *Figure 6A*).

We then asked whether the tissue specificities of the expression levels of target genes of the HOT loci reflect their involvement in the regulation of HK genes. For this purpose, we used the *tau* metric as reported by *Palmer et al., 2021*, where a high *tau* score (between 0 and 1) indicates a tissue-specific expression of a gene, whereas a low *tau* score means that the transcript is expressed stably across tissues. We observed that the average *tau* scores of target genes of HOT enhancers are significantly but by a small margin greater than the regular enhancers (0.66 and 0.63, respectively; p-value<$10^{-18}$, Mann-Whitney U test), with super-enhancers being equal to regular enhancers (0.63). The difference in the average *tau* scores of the HOT and regular promoters is stark (0.57 and 0.74, respectively, p-value<$10^{-100}$, Mann-Whitney U test), representing a 23% increase (*Figure 6B*). Combined with the involvement in the regulation of HK genes, average *tau* scores suggest that the HOT promoters are more ubiquitous than the regular promoters whereas HOT enhancers are more tissue-specific than the regular and super-enhancers. Further supporting this, the GO enrichment analysis showed that the GO terms associated with the set of genes regulated by HOT promoters are basic HK cellular functions (such as *RNA processing*, *RNA metabolism*, *ribosome biogenesis*, etc.), whereas HOT enhancers are enriched in GO terms of cellular response to the environment and liver-specific processes (such as *response to insulin, oxidative stress, epidermal growth factors,* etc.) (*Figure 6C*).

## A core set of HOT loci is active during development which expands after differentiation

Having observed that the HOT loci are active regions in many other human cell types, we asked if the observations made on the HOT loci of differentiated cell lines also hold true in the embryonic stage. To that end, we analyzed the HOT loci in H1 cells. It is important to note that the number of available

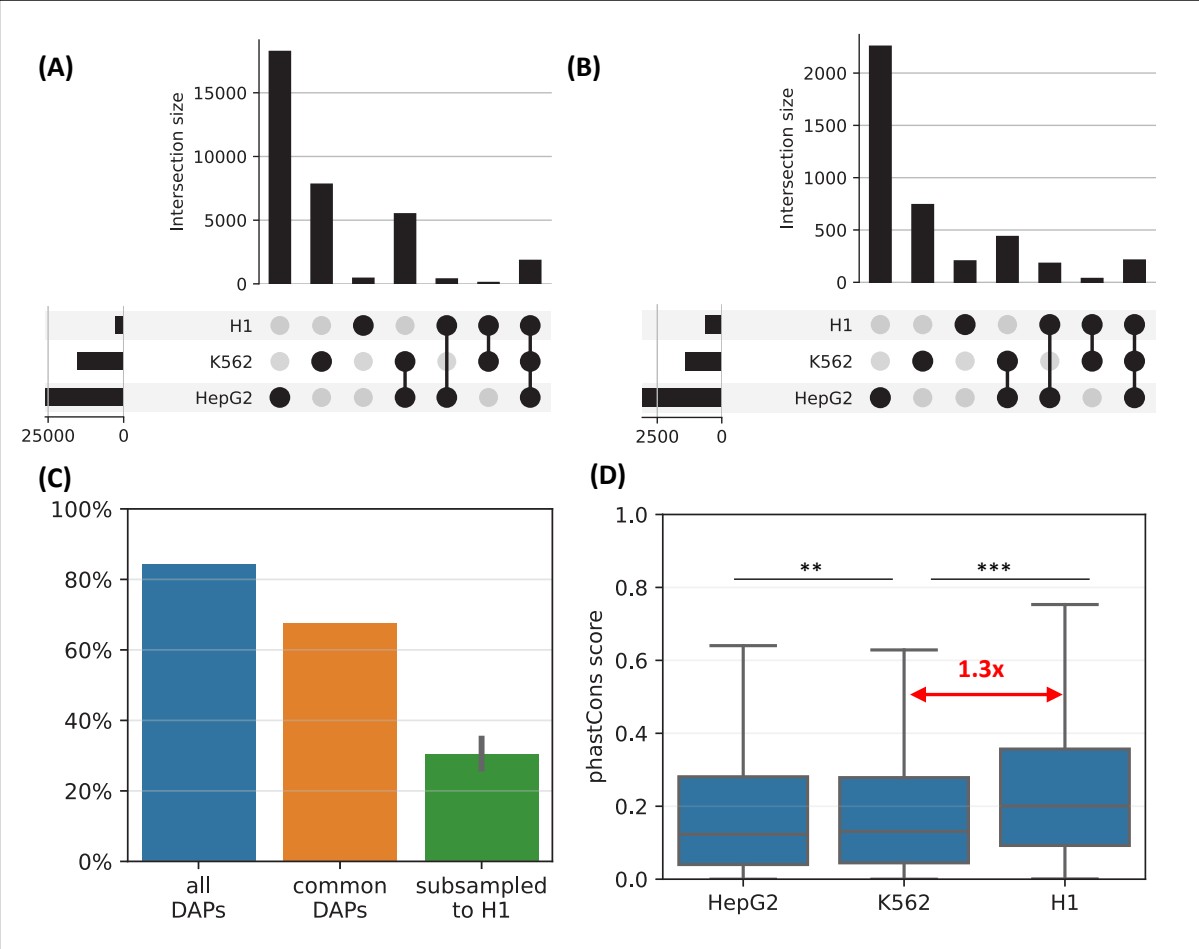

**Figure 7.** H1-hESC high-occupancy target (HOT) loci. (**A**) Overlaps between the HOT loci of three cell lines. (**B**) Overlaps between the HOT loci of cell lines defined using the set of DNA-associated proteins (DAPs) available in all three cell lines. (**C**) Fractions of H1 HOT loci overlapping with that of the HepG2 and K562 using the complete set of DAPs, common DAPs, and DAPs randomly subsampled in HepG2/K562 to match the size of H1 DAPs set. (**D**) phastCons scores of HOT loci in HepG2, K562, and H1.

The online version of this article includes the following figure supplement(s) for figure 7:

**Figure supplement 1.** GO terms associated with the high-occupancy target (HOT) enhancers and promoters in H1-hESC.

DAPs in H1 cells is significantly smaller (n=47) than in HepG2 and K562, due to a much smaller size of the ChIP-seq dataset generated in H1. Therefore, the criterion of having >17% of available DAPs yields n>15 DAPs for the H1, as opposed to 77 and 55 for HepG2 and K562, respectively. However, many of the features of the loci that we've analyzed so far demonstrated similar patterns (GC contents, target gene expressions, ChIP-seq signal values, etc.) when compared to the DAP-bound loci in HepG2 and K562, suggesting that albeit limited, the distribution of the DAPs in H1 likely reflects the true distribution of HOT loci. To alleviate the difference in available DAPs, in addition to comparing the HOT loci defined using the complete set of DAPs, we also (a) applied the HOT classification routing using a set of DAPs (n=30) available in all three cell lines, (b) randomly subselected DAPs in HepG2 and K562 to match the number of DAPs in H1.

We observed that, when the complete set of DAPs is used, 85% of the HOT loci of H1 are also HOT loci in either of the other two differentiated cell lines (*Figure 7A*). However, only <10% of the HOT loci of the two differentiated cell lines overlapped with H1 HOT loci, suggesting that the majority of the HOT loci are acquired after the differentiation. A similar overlap ratio was observed based on DAPs common to all three cell lines (*Figure 7B*), where 68% of H1 HOT loci overlapped with that of the differentiated cell lines. These overlap levels were much higher than the randomly selected DAPs matching the H1 set (30%, *Figure 7C*).

Average evolutionary conservation scores (phastCons) of the developmental HOT loci are 1.3× higher than K562 and HepG2 HOT loci (p-value<10⁻¹⁰, Mann-Whitney U test, *Figure 7D*). It is conceivable to hypothesize that the embryonic HOT loci are located mainly in regions with higher conservation regions, and more regulatory regions emerge as HOT loci after the differentiation. Some of these tissue-specific HOT loci could be those that are acquired more recently (compared to the H1 HOT loci), as it is known that the enhancers are often subject to higher rates of evolutionary turnover than the promoters (*Domené et al., 2013*).

GO enrichment analysis showed that H1 HOT promoters, similarly to the other cell lines, regulate the basic HK processes (*Figure 7—figure supplement 1*) while the HOT enhancers regulate responses to environmental stimuli and processes active during the embryonic stage such as *TORC1 signaling* and *beta-catenin-TCF assembly*. This suggests that the main processes that the HOT promoters are involved in during the development remain relatively unchanged after the differentiation (in terms of associated GO terms, and due to being the same loci as the HOT promoters in differentiated cell lines), whereas the scope of the cellular activities regulated by HOT enhancers gets expanded after differentiation to be more exclusively tissue-specific.

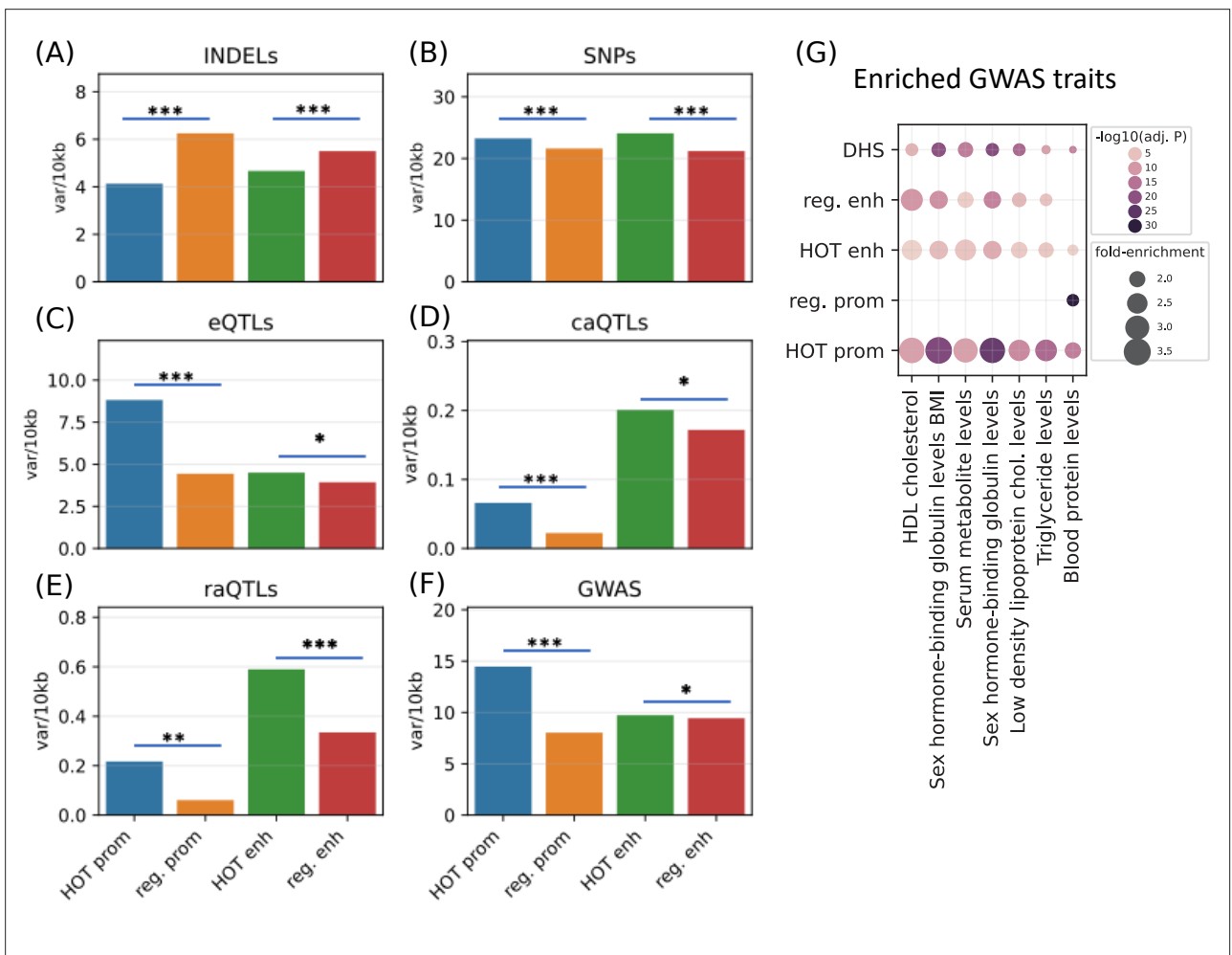

**Figure 8.** Densities of variants. (**A**) Common INDELs (MAF>5%), (**B**) common SNPs (MAF >5%), (**C**) eQTLs, (**D**) chromatin accessibility QTLs (caQTLs), (**E**) reporter array QTLs (raQTLs), and (**F**) GWAS and LD (r2>0.8) variants in high-occupancy target (HOT) loci and regular promoters and enhancers. (**G**) Enriched GWAS traits in HOT enhancers and promoters. All of the visualized data is generated from the HepG2 cell line.

The online version of this article includes the following figure supplement(s) for figure 8:

**Figure supplement 1.** GWAS traits enrichment analysis filtered by unadjusted p-values (p-value<0.001).

## HOT loci are enriched in causal variants

After establishing the expression and tissue specificities of the HOT loci, we next analyzed the polymorphic variability in HOT loci and whether these loci are enriched in phenotypically causal variants. First, we analyzed the density of common variants extracted from the gnomAD database (*Karczewski et al., 2020*) (filtered with MAF>5%). We observed that HOT enhancers and HOT promoters are depleted in INDELs (4.7 and 4.1 variants per 1 kb, respectively), compared to the regular enhancers and regular promoters (5.5 and 6.2 variants per 1 kb, p-value<$10^{-4}$ and <$10^{-100}$, respectively, Mann-Whitney U test; *Figure 8A*). Contradicting the pattern of conservation scores described above, the distribution of common SNPs is elevated in HOT enhancers and HOT promoters compared to regular enhancers and regular promoters (1.14× and 1.07× fold-enrichment, p-values<$10^{-20}$ and <$10^{-100}$, respectively, Mann-Whitney U test; *Figure 8B*). This elevation of common variants in HOT loci, despite being located in conserved loci, has been reported in a previous study in which the binding motifs of TFs were observed to colocalize in regions where the density of common variants was higher than average (*Vierstra et al., 2020*).

The eQTLs, on the other hand, are 2.0× enriched in HOT promoters compared to the regular promoters (p-value<$10^{-21}$, Mann-Whitney U test), while HOT enhancers are only moderately enriched in eQTLs compared to the regular enhancers (1.15×, p-value>0.05, Mann-Whitney U test; *Figure 8C*). eQTL enrichment in HOT promoters and regular promoters (compared to HOT and regular enhancers, respectively) is in line with the known characteristics of the eQTL dataset, that the eQTLs most commonly reflect TSS-proximal gene-variant relationships, and therefore are enriched in promoter regions since the TSS-distal eQTLs are hard to detect due to the burden of multiple tests (*Consortium, 2015*).

Unlike the eQTL analysis, we observed that the chromatin accessibility QTLs (caQTLs) are dramatically enriched in the overall enhancer regions (HOT and regular) compared to the promoters (HOT and regular) (4.1×, p-value<$10^{-100}$; Mann-Whitney U test, *Figure 8D*). This observation confirms the findings of the study which reported the caQTL dataset in HepG2 cells (*Currin et al., 2021*), which reported that the likely causal caQTLs are predominantly the variants disrupting the binding motifs of liver-expressed TFs enriched in liver enhancers. However, within the promoters regions, the HOT promoters are 3.0× enriched in caQTLs compared to the regular promoters (p-value=0.001; Mann-Whitney U test), whereas the fold enrichment in HOT enhancers is insignificant (1.2×, p-value=0.22, Mann-Whitney U test).

A similar enrichment pattern displays the reporter array QTLs (raQTLs; *van Arensbergen et al., 2019*), with respect to the overall (HOT and regular) promoter and enhancer regions, with 3.3× enrichment in enhancers (p-value<$10^{-10}$, Mann-Whitney U test, *Figure 8E*). But, within-promoters and within-enhancers enrichments show that the enrichment in HOT promoters is more pronounced than the HOT enhancers (3.6× and 1.8×, p-values<0.01 and<$10^{-11}$, respectively, Mann-Whitney U test). The enrichment of the raQTLs in enhancers over the promoters likely reflects the fact that the SNP-containing loci are first filtered for raQTL detection according to their capacities to function as enhancers in the reporter array (*van Arensbergen et al., 2019*).

Combined, all three QTL datasets show a pronounced enrichment in HOT promoters compared to the regular promoters, whereas only the raQTLs show significant enrichment in HOT enhancers. This suggests that the individual DAP ChIP-seq peaks in HOT promoters are more likely to have consequential effects on promoter activity if altered, while HOT enhancers are less susceptible to mutations. Additionally, it is noteworthy that only the raQTLs are the causal variants, whereas e/caQTLs are correlative quantities subject to the effects of LD.

Finally, we used the GWAS SNPs combined with the LD SNPs (r2>0.8) and observed that the HOT promoters are significantly enriched in GWAS variants (1.8×, p-value>$10^{-100}$) whereas the HOT enhancers show no significant enrichment over regular enhancers (p-value>0.1, Mann-Whitney U test) (*Figure 8F*). We then calculated the fold-enrichment levels of GWAS traits SNPs using the combined DHS regions of Roadmap Epigenome cell lines as a background (see Methods). Filtering the traits with significant enrichment in HOT loci (p-value<0.001, Binomial test, Bonferroni corrected, see Methods) left seven traits, of which all are definitively related to the liver functions (*Figure 8G*). Of the seven traits, only one (*Blood protein level*) was significantly enriched in regular promoters. While the regular enhancers are enriched in most of the (six of seven) traits, the overall enrichment values in HOT enhancers are 1.3× greater compared to the regular enhancers. The fold-increase is even greater

(1.5×) between the HOT and DHS regions. When the enrichment significance levels are selected using unadjusted p-values, we obtained 24 GWAS traits, of which 22 are related to liver functions (*Figure 8—figure supplement 1*). This analysis demonstrated that the HOT loci are important for phenotypic homeostasis.

## Transcriptional condensates as a model for explaining the HOT regions

Recent studies on phase-separated condensates have established that condensates are ubiquitous in cells and play crucial roles in gene regulation through transcriptional condensates (*Nair et al., 2019*; *Lee et al., 2022*; *Feric and Misteli, 2022*; *Ahn et al., 2021*). We postulated that the HOT loci could be explainable if it can be shown that the HOT loci demonstrate a high propensity for the formation of transcriptional condensates. The hallmarks of transcriptional condensates include (not limited to) scaffolding proteins that undergo liquid-to-liquid phase separation (LLPS), DNA and RNA molecules, and intrinsically disordered (IDR) proteins. We sought to analyze whether these properties can be attributed to the HOT loci.

First, using CD-CODE database (*Rostam et al., 2023*) we annotated 24% of the DAPs used in the analysis as LLPS-inducing proteins (*Figure 9A*). We observed that LLPS proteins are uniformly distributed in HOT loci (*Figure 9B*). We calculated a null distribution by randomly shuffling the ChIP-seq peaks in HOT loci 10 times, which resulted in a near-zero fraction of LLPS proteins located in >45% of the HOT loci, where the actual observed fraction is 23% (average of the last two bins in *Figure 9B*), strongly suggesting an overrepresentation. Moreover, LLPS proteins yield significantly stronger ChIP-seq signals compared to the rest of the DAPs (*Figure 9C*, p-value=0.002, t-test), and contain a higher percentage of predicted IDR regions (*Figure 9D*, 30% vs. 26%, p-value=0.01, t-test).

Next, we sought to quantify the RNA-related interactions in HOT loci. First, we used ENCODE's set of ChIP-seq datasets extracted using RNA-binding proteins (RBP) and observed that RBPs are more enriched in HOT loci compared to the rest of the DAPs in terms of fold-increase using ATAC-seq regions as background (*Figure 9E*, 1.5 vs. 1.3 in log2(FC), p-value=0.04, t-test). Second, we quantified the level of transcription using FANTOM, PINTS (*Yao et al., 2022*) (a modern tool for annotating eRNAs combining multiple types of RNA sequencing assays), and CAGE-seq peaks. We observed that all three types of annotations demonstrate high overrepresentation in HOT loci compared to regular promoters and enhancers by a factor of 2.7× on average (*Figure 9F*). Lastly, we used eCLIP datasets of 103 RBSs from the ENCODE Project and calculated the levels of RBP-RNA interactions. We observed that the difference in the levels of eCLIP signals in HOT loci and coding sequences are insignificant (1.31 vs. 1.4 in log2(FC), p-value=0.4, t-test), while in regular promoter and enhancer regions, the eCLIP signals are depleted compared to the ATAC-seq regions with the log2(FC) values of –0.1 and –0.05, respectively (p-value<$10^{-30}$, t-test), suggesting a strong RNA-related component in the composition of 3D medium surrounding the HOT loci.

All this data suggests a strong likelihood of involvement of transcriptional condensates in the mechanisms leading to the phenomena of HOT loci.

## Discussion

HOT loci have been noticed and studied in different species since the early years of the advent of the ChIP-seq datasets (*Roy et al., 2010*; *Moorman et al., 2006*; *Gerstein et al., 2010*; *Kvon et al., 2012*; *Yip et al., 2012*; *Xie et al., 2013*). Up until recently, most of the studies have extensively studied the reasons through which the ChIP-seq peaks appeared to be binding to HOT loci and characterized certain sequence features of the HOT loci which could enable elevated read mapping rates (*Moorman et al., 2006*; *Teytelman et al., 2013*; *Wreczycka et al., 2019*). As the number of assayed DAPs in multiple human cell types and model organisms has increased, however, the assumption of the HOT loci being exceptional cases and results of false positives in ChIP-seq protocols have given way to the acceptance that the HOT loci, with exorbitant numbers of mapped TFBSs, are indeed hyperactive loci with distinct features characteristic of active regulatory regions (*Ramaker et al., 2020*; *Partridge et al., 2020*).

In this study, we studied the HOT loci in multiple complementary aspects to the previous works and expanded the scope of characterization extensively using the functional genomics datasets. We used the two most extensively characterized differentiated cell lines of the ENCODE Project: HepG2

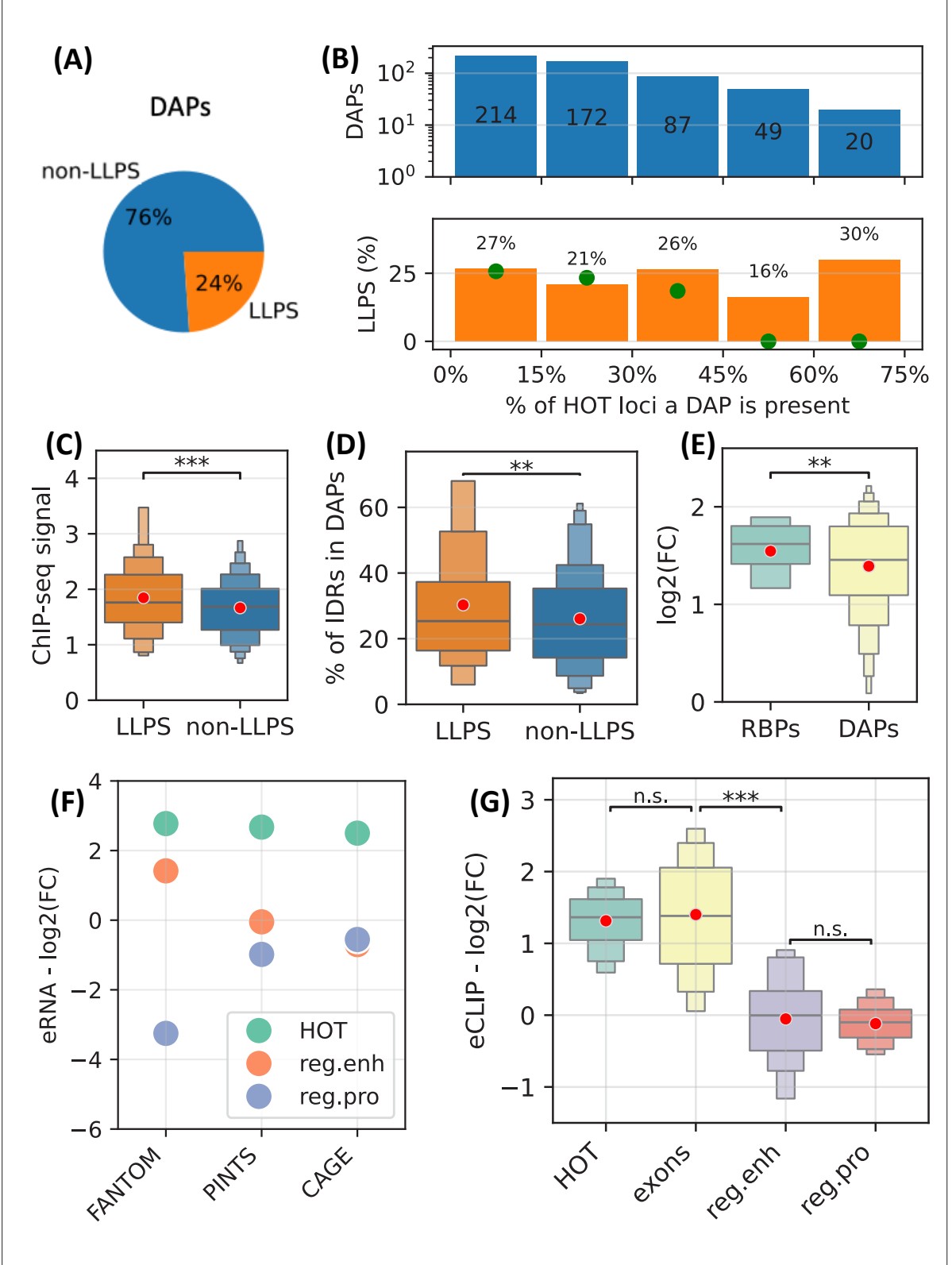

**Figure 9.** High-occupancy target (HOT) loci as transcriptional condensates. (**A**) Fraction of DNA-associated proteins (DAPs) annotated as liquid-to-liquid phase separation (LLPS) proteins in CD-CODE database. (**B**) (Upper) Distribution of DAPs in HOT loci binned by the % of HOT loci they overlap with. (Lower) % of DAPs in the bins annotated as LLPS. Green points are the expected percentage values obtained by randomly shuffling the peaks in HOT loci 10 times. (**C**) Z-scores of ChIP-seq signal values of LLPS proteins and the rest of the DAPs in HOT loci. (**D**) % of the protein lengths predicted

*Figure 9 continued on next page*

*Figure 9 continued*

as IDRs (MobiDB) in LLPS proteins and the rest of the DAPs. (**E**) Enrichment of ChIP-seq peaks of RNA-binding proteins (RBP) and the rest of the DAPs. (**F**) Enrichment of FANTOM, PINTS, and CAGE regions in HOT, regular enhancers, and regular promoters. (**G**) Enrichment of eCLIP RBP-RNA interactions in HOT, exons, regular enhancers, and regular promoters. (**E–G**) Enrichment values are quantified as log2(fold-change) with ATAC-seq regions as a background. (**C–E, G**) Red dots represent the mean values of the boxplots.

and K562. We also included the H1-hESC human stem cells to study the activities of HOT loci during the embryonic stage. The number of assayed DAPs in these cell lines is far from complete (*Lambert et al., 2018*), therefore it is important to note that as the sizes of the assayed DAP ChIP-seq datasets increase, our understanding of the mechanisms of HOT loci will certainly improve. However, the core principles can already be inferred using the currently available datasets. Previous studies have used different metrics to define the HOT loci. For example, *Wreczycka et al., 2019*, used the 99th percentile of the density of TFBSs for a 500 bp sliding window, *Ramaker et al., 2020*, used the window length of 2 kb and required >25% of TFs to be mapped, *Partridge et al., 2020*, used loci with >70 chromatin-associated proteins in 2 kb window. These heterogeneous definitions, however, fail to appreciate that the histogram of loci binned by the number of harbored TFBSs represents an exponential distribution (*Figure 1A*). We, therefore, applied our analyses both to the binarily defined HOT and non-HOT loci, as well as to the overall spectrum of loci in the context of TFBS density. This approach allowed us to better understand the correlations of characteristics of loci with the TF activity. Noticeably, this approach showed us that the HOT loci have their propensities to engage in long-range chromatin contacts with other equally or more DAP-bound loci than less active ones, making it more clear that the HOT loci are located in 3D hubs and FIREs (*Figure 3A*).

Using the datasets generated in H1 we established that only <10% of the HOT loci in two differentiated cell lines overlap with the HOT loci of stem cells. This points to the high tissue specificity of the HOT loci. Previous studies have also concluded that the HOT loci are not constitutive by nature, and are established in a dynamic manner after the differentiation (*Boyle et al., 2014*).

Previous studies have carried out extensive mapping of the known binding motifs of TFs to the HOT loci and identified a small set of 'anchor' binding motifs of a few key tissue-specific TFs (*Moorman et al., 2006*; *Ramaker et al., 2020*), and proposed that perhaps these driver TFs initiated the formation of HOT loci, similar to how the pioneer factors function. Other studies have concluded that the vast majority of the peaks do not contain the corresponding motifs and that most of the mapped peaks represent indirect binding through TF-TF interactions (*Ramaker et al., 2020*; *Partridge et al., 2020*; *Vierstra et al., 2020*; *White et al., 2021*). We relied on these studies and focused on aspects of the HOT loci other than the quantification of known binding motifs of DAPs in HOT loci. Interestingly, the high prediction accuracy of our deep learning model is in agreement with the notion of the existence of shared motifs among the HOT loci but also implies that the indirectly bound loci also carry shared sequence features, perhaps other than the binding motifs or weak motifs which are not detected using the traditional PWM-based tools of motif detection.

Another model that has been increasingly attributed to the formation and maintenance of long-range 3D chromatin interactions involves phase-separated condensates (*Nair et al., 2019*; *Lee et al., 2022*; *Feric and Misteli, 2022*; *Ahn et al., 2021*). Some enhancers were shown to drive the formation of large chromosomal assemblies involving a high concentration of TFs (*Nair et al., 2019*). In general, it has been increasingly appreciated that condensates ubiquitously attract and activate enhancers (*Shrinivas et al., 2019*; *Wei et al., 2020*; *Boija et al., 2018*). The detection of condensates relies on low-throughput live-cell imaging methods such as FISH, which often involves only a few tagged molecules. Therefore, currently, to the best of our knowledge, there are no datasets of condensate formation with large numbers of molecules simultaneously that we could use to draw statistical conclusions. However, there is already an increasing body of research reporting on the characteristic hallmarks that the transcriptional condensates share (*Palacio and Taatjes, 2022*; *Mitrea et al., 2022*; *Gelder et al., 2024*; *Bhat et al., 2021*; *Rippe and Papantonis, 2021*). We used those hallmarks as telltale signs and made a case for the likelihood of the HOT loci being sites with a high propensity of forming condensates. A condensate can start forming with only one bound TF and a cofactor, e.g. OCT4 and Mediator (*Shrinivas et al., 2019*), which requires the presence of a strong binding motif of the condensate-initiating TF. Once the condensates of sufficient size form, the kinetic trap that it creates can facilitate the accumulation of a soup of DAPs,

which then can undergo high-intensity protein-protein and protein-DNA and protein-RNA interactions, many constituents of which then get mapped to the involved DNA regions upon ChIP-seq experiments. This model can incorporate the seemingly contradictory conclusions of (a) the vast majority of DAPs lacking the binding motifs in HOT loci and (b) a high accuracy of sequence-based classification of HOT loci using the CNN models. It is important to note here that our proposed condensate model is a speculative hypothesis. Further experimental studies in the field are needed to confirm or reject it.

One of the main limitations of our study is the lack of higher-resolution TF-DNA interaction datasets such as CUT&RUN, ChIP-exo, or single-cell versions of the assets used in this study. Furthermore, one of the hallmarks of condensates is the overrepresentation of certain structural motifs in LLPS proteins, which we did not pursue due to size limitations. Further studies addressing these topics hold promise to shed more light on the subject of HOT loci.

## Methods
### Datasets

TF (DAP), histone modification, DHS ChIP-seq, and ATAC-seq datasets for HepG2, K562, H1-hESC cell lines were batch downloaded from the ENCODE Project (*Wang et al., 2013*). For each DAP of each cell line, if there were multiple datasets, the one with the latest date was selected, prioritizing the ones with the least among the audit errors and warnings (*Supplementary file 1, table S1*). The GRCh37/hg19 assembly was used as a reference genome throughout the study. In those cases when ChIP-seq dataset was reported on GRCh38/hg38, the coordinates were converted to hg19 using liftOver. The phastCons evolutionary conservation scores generated from 46 vertebrate species, placental mammals, and primates. For comparing, averaged values of phastCons scores over the 400 bp loci were used. CpG islands, repeat elements, and GENCODE TSS annotations were all obtained from the UCSC genome browser database (*Davis et al., 2018*). Transcribed enhancer regions (eRNAs) were obtained from the FANTOM database (*Lizio et al., 2019*). Super-enhancer regions were obtained from *Hnisz et al., 2013*.

Hi-C datasets were obtained from ENCODE Project. See Appendix 1 – Hi-C 3D chromatin analysis for a detailed description of Hi-C data analysis.

GC contents were calculated using the 'nuc'' functionality of the bedtools program (*Quinlan and Hall, 2010*). Gene expression data was obtained from the Roadmap Epigenomics Project. For analyzing the expression levels of target genes, the gene of the overlapping TSS was used for promoters, whereas for enhancers, the nearest genes were selected using the *bedtools closest* function. Tissue specificity metric *tau* scores for genes were downloaded from *Palmer et al., 2021*.

LLPS protein annotations were obtained from CD-CODE website https://cd-code.org. Predicted intrinsically disordered region annotations of proteins were obtained from MobiDB website https://mobidb.org. RBP ChIP-seq datasets used in the study are in *Supplementary file 1, table S6*. eCLIP datasets used in the study are in *Supplementary file 1, table S7*. PINTS eRNA dataset was obtained from https://pints.yulab.org. CAGE datasets were downloaded from ENCODE (ENCFF184VBV, ENCFF246WDH, ENCFF933JJT) and merged.

**Table 1.** Schema of classifying loci according to the number of bound DNA-associated proteins (DAPs).
The initial 4 bins are loci bound by DAPs increasing linearly from 1 to 5 (gray fields). The remaining 10 bins are defined by edge values increasing on a logarithmic scale from 5 to the maximum number of available DAPs in each cell line (orange and red fields) using the Numpy formula np.logspace(np.log10(5), np.log10(max_tfs), 11, dtype = int). HOT loci correspond to the last 5 bin edges (red fields).

| Bin edges (n=15) | | | | | | | | | | | | | | |
|---|---|---|---|---|---|---|---|---|---|---|---|---|---|---|
| HepG2 | 1 | 2 | 3 | 4 | 5 | 7 | 12 | 19 | 31 | 48 | 77 | 122 | 192 | 304 | 480 |
| K562 | 1 | 2 | 3 | 4 | 5 | 7 | 11 | 16 | 24 | 37 | 55 | 82 | 123 | 184 | 275 |
| H1 | 1 | 2 | 3 | 4 | 5 | 6 | 7 | 8 | 10 | 12 | 15 | 18 | 22 | 26 | 32 |
| | Linear growth (n=4) | | | | | Logarithmic growth (n=10) | | | | | | | | | |

## Definitions

The loci were divided into bins according to a two-part scale. The first part is on a linear scale from 1 to 5 (4 bins), the second part is on a natural logarithmic scale from 5 to the maximum number of DAPs bound to a single locus in that cell line (10 bins) (*Table 1*).

We considered an average TFBS to be 8 bp long (*Vinson et al., 2011*; *Wunderlich and Mirny, 2009*). Given that we analyzed the loci in 400 bp, we reasoned that, theoretically, there can be at most 50 simultaneous binding events in the locus (8×50 = 400). Therefore, we considered the bins containing >50 DAPs in K562 as HOT loci, which meant the last four bins in *Table 1*. The reason we chose K562 for setting the threshold was the fact that K562 is the lesser of the two most TF ChIP-seq abundant cell lines. So, the corresponding threshold number for HepG2 is >77 TFs.

These nominal numbers are used in cases when the distributions are displayed for individual cell lines (such as *Figure 1A* and *Figure 1—figure supplement 1*). When the figures display the distributions for two cell lines in a joint manner (such as *Figure 3A and B*), the edges are converted to the average percentages of the overall scale lengths for each cell line.

*Regular enhancers* were defined as central 400 bp regions of DHS which overlap H3K27ac histone modification regions with promoter and exons removed from them.

*Promoters* were defined as 1.5 kb upstream and 500 bp downstream regions of the canonical and alternative TSS coordinates were extracted from the knownGenes.txt table obtained from UCSC Genome Browser.

All the genomic arithmetic operations were done using the *bedtools* program (*Quinlan and Hall, 2010*). Figures were generated using Matplotlib (*Hunter, 2007*) and Seaborn (*Waskom, 2021*) packages. Statistical and numerical analyses were done using the pandas, *NumPy*, *SciPy*, and *sklearn* packages (*Virtanen et al., 2020*) in *Python* programming language. Genomic repeat regions were extracted from *RepeatMasker* table obtained from http://www.repeatmasker.org/. CpG islands were extracted from *cpgIslandExt* table obtained from the UCSC Genome Browser. Protein-protein interaction network information was obtained using the https://string-db.org web interface (*Szklarczyk et al., 2019*).

## Statistical analyses

All the statistical significance analyses were done using the *SciPy* package. Statistical significance of genomic region overlaps was calculated using the '*bedtools fisher*' command. The p-values too small to be represented by the command line output were represented as $<10^{-100}$.

Correlation values with the number of bound TFs were calculated using the average of the value for the bins, and the midpoint numbers of the edges of each bin.

For calculating the statistical significance, we used the non-parametric Mann-Whitney U test when the compared data points are non-linearly correlated and multi-modal. When the data distributions are bell-curve shaped, the Student's t-test was used.

## GWAS analysis

NHGRI-EBI GWAS database variants were grouped according to their traits (dataset e0_r2022-11-29). For each GWAS SNP, LD SNPs with r2>0.8 were added using the *plink v1.9* (*Chang et al., 2015*) program using the parameters `--ld-window-r2 0.8 --ld-window-kb 100 --ld-window 1000000`. Enrichments of GWAS-trait SNPs were calculated as the ratios of densities of SNPs in each class of regions (e.g. HOT enhancers, HOT promoters) to either that of the regular enhancers or the DHS regions. Statistical significance of enrichment was calculated using the binomial test. FDR values were calculated using the Bonferroni correction.

## Sequence classification analysis

Classification tasks were constructed in a binary classification setup. The control regions were used from: (a) randomly selected (10× the size of the HOT loci) merged DHS regions from all the available datasets from Roadmap Epigenomic Project, (b) all of the promoter regions as defined above, (c) regular enhancers as defined above, with the HOT loci subtracted (see Appendix 1 – Classification datasets for details).

## Sequence-based classification (CNN)

Sequences were converted to one-hot encoding and a CNN was trained using each of the control regions as negative set. The model was built using *tensorflow v2.3.1* (*Abadi et al., 2016*) and trained on NVIDIA k80 GPUs (see Appendix 1 – Sequence-based classification for details).

## Sequence-based classification (SVM)

SVM models were trained using the LS-GKM package (*Lee, 2016*) (see Appendix 1 – Sequence-based classification for details).

## Feature-based classification

Sequences were represented in terms of GC, CpG, GpC contents and overlap percentages with annotated CpG islands. SVM classifiers were trained using these sequence features (see Appendix 1 – Feature-based classification for details).

## Variant analysis

Common SNPs and INDELs were extracted from the *gnomAD r2.1.1* dataset (*Karczewski et al., 2020*). Variants with PASS filter value and MAF>5% were selected using the "view -f PASS -i 'MAF[0]>0.05'" options of *bcftools* program (*Li, 2011*). Loss-of-function variants were downloaded from the *gnomAD* website under the option 'all homozygous LoF curation' section of v2.1.1 database. raQTLs were downloaded from https://sure.nki.nl (*van Arensbergen et al., 2019*). Liver and blood eQTLs were extracted from the GTEx v8 dataset (https://www.gtexportal.org/home/datasets). Liver caQTLs were obtained from the supplementary material of *Currin et al., 2021*. NHGRI-EBI GWAS database variants were grouped according to their traits (dataset e0_r2022-11-29). For each GWAS SNP, LD SNPs with r2>0.8 were added using the *plink v1.9* program using the parameters '--ld-window-r2 0.8 --ld-window-kb 100 --ld-window 1000000'. Enrichments of GWAS-trait SNPs were calculated as the ratios of densities of SNPs in each class of regions (e.g. HOT enhancers, HOT promoters) to either that of the regular enhancers or the DHS regions. The statistical significance of enrichment was calculated using the binomial test. FDR values were calculated using the Bonferroni correction.

## Acknowledgements

This work utilized the computational resources of the NIH HPC Biowulf cluster (http://hpc.nih.gov). This research was supported by the Intramural Research Program of the National Library of Medicine, National Institutes of Health.

## Additional information

### Funding

| Funder | Grant reference number | Author |
| --- | --- | --- |
| National Institutes of Health | | Sanjarbek Hudaiberdiev Ivan Ovcharenko |

The funders had no role in study design, data collection and interpretation, or the decision to submit the work for publication.

### Author contributions

Sanjarbek Hudaiberdiev, Conceptualization, Visualization, Methodology, Writing – original draft, Data curation, Formal analysis, Investigation, Resources, Software, Validation; Ivan Ovcharenko, Conceptualization, Formal analysis, Supervision, Project administration, Writing – review and editing, Investigation, Validation

### Author ORCIDs

Sanjarbek Hudaiberdiev https://orcid.org/0000-0003-0860-5250

Reviewer #1 (Public review): https://doi.org/10.7554/eLife.95170.3.sa1
Reviewer #2 (Public review): https://doi.org/10.7554/eLife.95170.3.sa2
Reviewer #3 (Public review): https://doi.org/10.7554/eLife.95170.3.sa3
Author response https://doi.org/10.7554/eLife.95170.3.sa4

## Additional files

### Supplementary files

• MDAR checklist

• Supplementary file 1. Supplementary tables. Columns are explained in each sheet. S1: List of ENCODE ChIP-seq datasets used in the study. S2: Coordinates of high-occupancy target (HOT) loci defined in three cell lines. S3: List of DNA-associated proteins (DAPs) clustered into four groups and their PPI enrichment summary statistics. S4: List of ultraconserved regions overlapping with HOT loci. S5: Comparison of HOT loci defined using ENCODE vs. Roadmap Epigenome Project datasets. S6: List of ChIP-seq datasets of RNA-binding protein used in the study. S7: List of eCLIP datasets of RNA-binding proteins used in the study.

### Data availability

All the used and produced data presented in this manuscript are deposited in Zenodo. The codebase used for generating the results presented in this manuscript is available at GitHub, copy archived at *Hudaiberdiev, 2024*.

The following dataset was generated:

| Author(s) | Year | Dataset title | Dataset URL | Database and Identifier |
|---|---|---|---|---|
| Hudaiberdiav S, Ovcharenko I | 2024 | Functional characteristics and computational model of abundant hyperactive loci in the human genome | https://zenodo.org/records/13271790 | Zenodo, 10.5281/zenodo.7845120 |

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

## Appendix 1

### Joint DAPs analysis

To jointly analyze the conditional distributions of ChIP-seq signal levels in the presence/absence of individual DAPs, we extracted a square matrix of size n=545 using the DAPs present in HepG2. For each analyzed 400 bp locus, we extracted bound DAPs together with the ChIP-seq signal values. Then, each binary combination of the DAPs bound in that locus is added to their respective cells on the square matrix. Afterward, the matrix is normalized along the x-axis with the maximum value of each row. In other words, each row represents the normalized ChIP-seq signal strength in the presence of DAP indicated on the x-axis. We treated the empty values as 0 and removed the main diagonal values, leading to the matrix size of 544×544. Hierarchical clustering was done using the UPGMA algorithm.

### Hi-C 3D chromatin analysis

Hi-C data analysis was carried out on HepG2. The datasets were obtained from ENCODE Project: ENCFF050EKS (chromatin loops), ENCFF018XKF (TADs), ENCFF548XLR (hic file). The coordinates in all of the datasets were converted from hg38 to hg19 using LiftOver.

The significant long-range contacts with 5 kb resolution were extracted using the FitHiChIP program (*Bhattacharyya et al., 2019*) with a threshold q-value<0.0001 and ICE bias correction, using *all-against-all* option.

The loci with >50% overlap were considered for the analysis of loops, TADs, and long-range chromatin contacts. Using the long-range chromatin contacts, we constructed a graph such that each node is the analyzed 400 bp locus and the edge is a long-range chromatin contact if the two connected nodes are located on different legs of the chromatin contacts. Based on this graph, we calculated the total number of contacts between the loci located in different bins, leading to a 14×14 matrix. We then normalized the values in each cell of the matrix with the maximum number of contacts in all cells.

FIRE loci were extracted from the *.hic* file using FIRECaller R package (*Schmitt et al., 2016*).

For enrichment analyses of all the mentioned Hi-C-related regions, the ATAC-seq regions were used as background.

### PPI enrichment analysis

To test the significance of the PPI networks described above, we ran 100 trials for each cluster by randomly selecting an equal number of DAPs reported in PPI networks and calculated the significance of the PPI enrichment p-values. All of the reported PPI enrichment p-values were significantly higher than the randomized trials (p-value<0.01, one-sample t-test).

PPI networks and PPI enrichment p-values were extracted using the STRING Database's API (https://string-db.org/cgi/help.pl?subpage=api). For each cluster of DAPs analyzed, we submitted the list of DAPs as identifiers and retrieved the p-values using the *ppi_enrichment* interface. For each cluster, we extracted 100 PPI enrichment p-values each time randomly selecting DAPs in equal numbers to the size of the analyzed cluster. We then used the set of 100 p-values as a background distribution and conducted a one-sample t-test, where by the null hypothesis the p-value of the cluster is the mean of 100 p-values and computed the p-values of significance of the reported PPI network. The results of this analysis are in *Supplementary file 1, table S2*.

### Classification results analyses

Sequence-based classification experiments were carried out using CNNs (one-hot encoded) and gkmSVM (gapped k-mers). For feature-based classification, we trained logistic regression (LogReg) classifiers and separate SVM classifiers using kernel functions of linear, polynomial, RBF, and sigmoid.

Using the sequence features, we trained separate models using each of the features in addition to one with all of the features combined. We observed that, when averaged across all the methods, GC content value possesses the highest amount of discrimination power (auROC: 0.73), followed by the combination of all features (auROC: 0.70) (*Appendix 1—figure 1A*). When compared across the classification methods, LogReg and SVM with linear kernel outperformed the other non-linear kernels by 20%, suggesting that the features possess linearly combined or largely overlapping effects in encoding the information in HOT loci (*Appendix 1—figure 2B*).

When classified using the sequences directly, CNN yielded the highest performance with auROC of 0.91, while for the gkmSVM it was 0.86 (both averaged over cell lines and control sets), suggesting that CNNs capture the motif grammar of the HOT loci better than gapped k-mers (*Appendix 1—figure 2*). When the two classification schemes (sequence- and feature-based) are compared, CNNs outperformed the LogReg and linear SVMs by a factor of 1.3× (or 17%).

## Classification datasets

For the classification of HOT loci, three different setups were constructed using the control (negative) sets:

- Randomly selected from the merged DHS regions obtained from the Roadmap Epigenomics Project to be 10× the size of the positive set (HOT loci)
- Regular enhancers (see Methods: Definitions), with the HOT loci subtracted
- Regular promoters (see Methods: Definitions)

The regions from chromosomes 6,7 were used as validation sets, chromosomes 8,9 were used as test sets, and the rest of the autosomal chromosomes were used as training sets.

The total number of regions in classification setups and their train/validation/test sets splits is as follows:

| Controls: DHS | | | | | |
|---|---|---|---|---|---|
| Cell line | HOTs | Controls | Train | Validation | Test |
| HepG2 | 25,928 | 249,499 | 210,520 | 33,231 | 31,676 |
| K562 | 15,231 | 146,585 | 123,041 | 20,310 | 18,465 |
| Controls: regular enhancers | | | | | |
| Cell line | HOTs | Controls | Train | Validation | Test |
| HepG2 | 25,928 | 249,499 | 210,520 | 33,231 | 31,676 |
| K562 | 15,231 | 146,585 | 123,041 | 20,310 | 18,465 |
| Controls: regular promoters | | | | | |
| Cell line | HOTs | Controls | Train | Validation | Test |
| HepG2 | 25,928 | 28,621 | 34,970 | 5479 | 3403 |
| K562 | 15,231 | 25,810 | 41,979 | 5800 | 3959 |

## Sequence-based classification

For training CNNs, the sequences of the loci were converted to one-hot encoding, with the lengths options of 400 bp and extended to 1000 bp.

The model consists of the layers as follows:

| Layer | Params | Activation |
|---|---|---|
| 1. Convolutional | filters = 480, kernel_size = 9, stride = 1 | ReLu |
| 2. Max pool | Pool_size = 9, stride = 3 | |
| 3. Droupout | p=0.2 | |
| 4. Convolutional | filters = 480, kernel_size = 4, stride = 1 | ReLu |
| 5. Max pool | Pool_size = 4, stride = 2 | |
| 6. Droupout | p=0.2 | |
| 7. Convolutional | filters = 240, kernel_size = 4, stride = 1 | ReLu |
| 8. Max pool | Pool_size = 4, stride = 2 | |
| 9. Droupout | p=0.2 | |
| 10. Convolutional | filters = 320, kernel_size = 4, stride = 1 | ReLu |

*Continued on next page*

*Continued*

| Layer | Params | Activation |
|---|---|---|
| 11. Max pool | Pool_size = 4, stride = 2 | |
| 12. Fully connected | units = 180 | ReLu |
| 13. Fully connected | units = 15 | Sigmoid |

Total number of trainable parameters is 2,342,723. The kernels were subjected to constraints of max_norm = 0.9, l1=5*10E-7, l2=1E-8. Each instance of the model was trained using the input lengths of 400 and 1000. The training process was run for a maximum of 200 epochs with a patience period of 15. The models were built using *tensorflow v2.3.1* and trained on NVIDIA k80 GPUs.

For SVM classification, gapped k-mer SVM program was used and downloaded from https://github.com/Dongwon-Lee/lsgkm (*Lee, 2023*). For each category of the regions, instances of SVM models were trained, using 400 bp and 1000 bp regions, with the following kernel options:

    0 -- gapped-kmer
    1 -- estimated l-mer with full filter
    2 -- estimated l-mer with truncated filter (gkm)
    3 -- gkm+RBF (gkmrbf)
    4 -- gkm+center weighted (wgkm)
    5 -- gkm+center weighted+RBF (wgkmrbf)

## Feature-based classification

The features used for classification were:

    - GC content.
    - CpG content: counted the occurrences of 'CG' as density over the sequence length.
    - GpC content: counted the occurrences of 'GC' as density over the sequence length.
    - CpG island coverage: fraction of the overlaps with the CpG island obtained from UCSC Genome Browser database.

Each classification model was trained using all of the features at once (n=4) and using each of the features separately.

### Logistic regression

*sklearn.linear_model.LogisticRegression* API was used from scikit-learn library.

### SVM

sklearn.svm.SVM API was used from scikit-learn library. Kernels used with SVM classification are *linear*, *polynomial*, *radial basis function(rbf)*, and *sigmoid*.

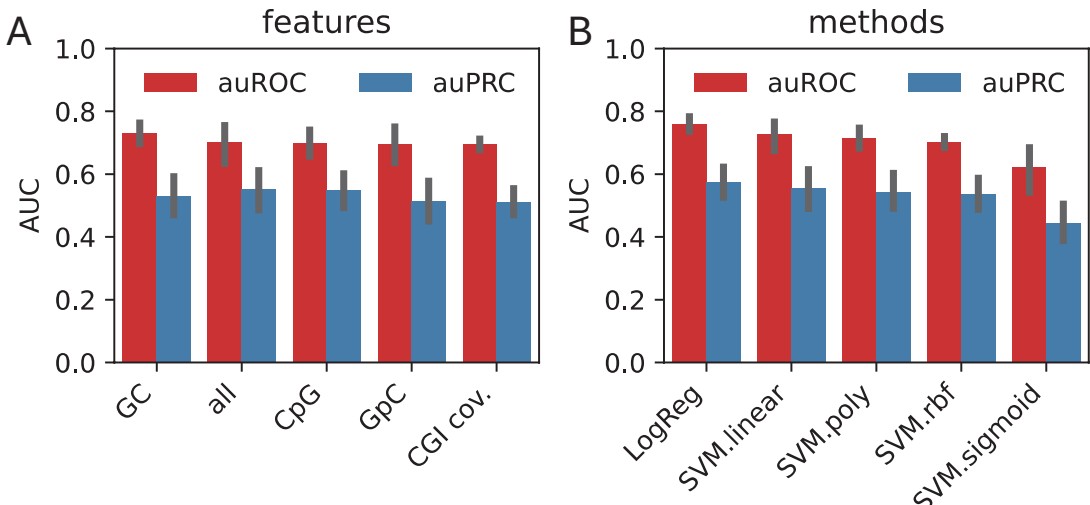

**Appendix 1—figure 1.** Classification of high-occupancy target (HOT) loci using the sequence features. (**A**) Classification performances when each sequence feature (*GC, GpC, CpG, CGI*) is used separately and all of them simultaneously (*all*). Error bar variations across cell lines, classification methods, and control sets. (**B**) Classification performances of different methods. Error bar variations across cell lines, sequence features, and control sets.

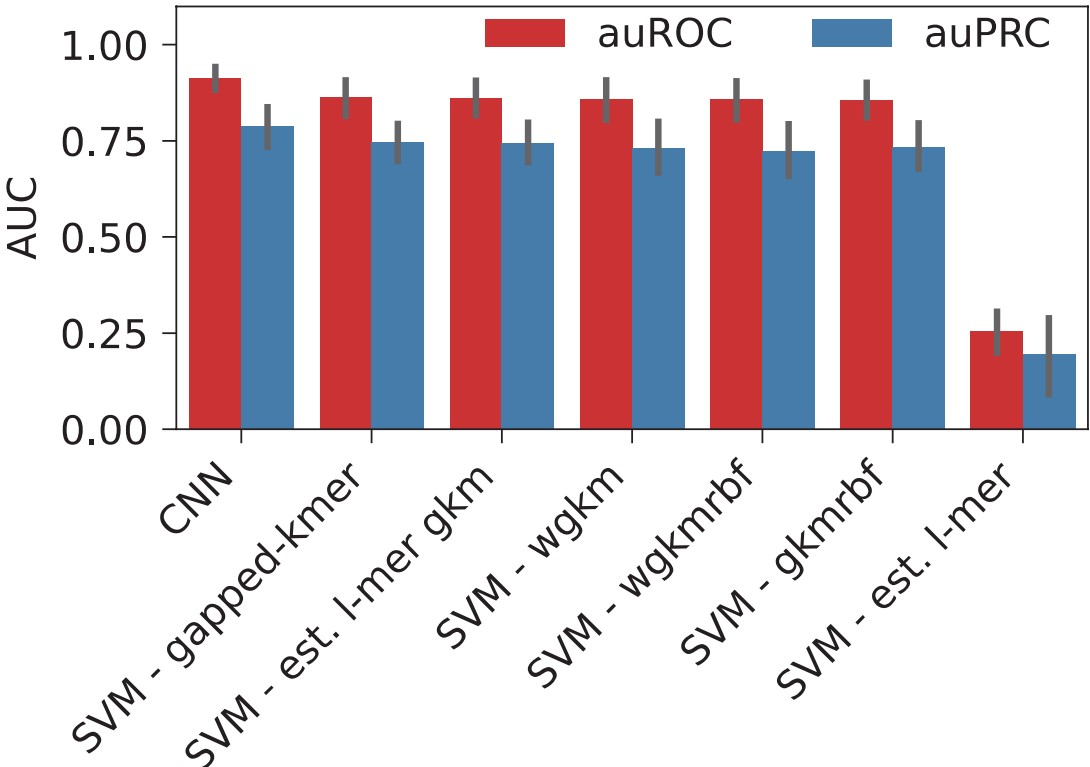

**Appendix 1—figure 2.** Classification of high-occupancy target (HOT) loci using the sequences directly. Error bar variations across cell lines and control sets. See Appendix 1 – Sequence-based classification for details of methods used.

