## [Editor Report · eLife Assessment]

This **valuable** study explores the sequence characteristics and conservation of high-occupancy target loci, regions in the human genome such as promoters and enhancers that are bound by a multitude of transcription factors. The computational analyses presented in this study are **solid**. This study would be a helpful resource for researchers performing ChIP-seq based analyses of transcription factor binding.

---

## [Referee Report · Reviewer #1 (Public review)]

Summary:

This study explores the sequence characteristics and features of high-occupancy target (HOT) loci across the human genome. The computational analyses presented in this paper provide information into the correlation of TF binding and regulatory networks at HOT loci that were regarded as lacking sequence specificity.

By leveraging hundreds of ChIP-seq datasets from the ENCODE Project to delineate HOT loci in HepG2, K562, and H1-hESC cells, the investigators identified the regulatory significance and participation in 3D chromatin interactions of HOT loci. Subsequent exploration focused on the interaction of DNA-associated proteins (DAPs) with HOT loci using computational models. The models established that the potential formation of HOT loci is likely embedded in their DNA sequences and is significantly influenced by GC contents. Further inquiry exposed contrasting roles of HOT loci in housekeeping and tissue-specific functions spanning various cell types, with distinctions between embryonic and differentiated states, including instances of polymorphic variability. The authors conclude with a speculative model that HOT loci serve as anchors where phase-separated transcriptional condensates form. The findings presented here open avenues for future research, encouraging more exploration of the functional implications of HOT loci.

Strengths:

The concept of using computational models to define characteristics of HOT loci is refreshing and allows researchers to take a different approach in identifying potential targets. The major strengths of the study lie in the very large number of datasets analyzed, with hundreds of ChIP-seq data sets for both HepG2 and K562 cells as part of the ENCODE project. Such quantitative power allowed the authors to delve deeply into HOT loci, which were previously thought to be artifacts.

Weaknesses:

While this study contributes to our knowledge of HOT loci, there are critical weaknesses that need to be addressed. There are questions on the validity of the assumptions made for certain analyses. The speculative nature of the proposed model involving transcriptional condensates needs either further validation or be toned down. Furthermore, some apparent contradictions exist among the main conclusions, and these either need to be better explained or corrected. Lastly, several figure panels could be better explained or described in the figure legends.

Update After Revisions:

The authors have addressed the above comments and concerns appropriately. The addition of the new Figure 9 is particularly compelling and strengthens the authors' conclusions. This reviewer has no further concerns.

---

## [Referee Report · Reviewer #2 (Public review)]

Summary:

The paper by Hydaiberdiev and Ovcharenko offers comprehensive analyses and insights about the 'high-occupancy target' (HOT) loci in the human genome. These are considered genomic regions that overlap with transcription factor binding sites. The authors provided very comprehensive analyses of the TF composition characteristics of these HOT loci. They showed that these HOT loci tend to overlap with annotated promoters and enhancers, GC-rich regions, open chromatin signals, and highly conserved regions and that these loci are also enriched with potentially causal variants with different traits.

Strengths:

Overall, the HOT loci' definition is clear and the data of HOT regions across the genome can be a useful dataset for studies that use HepG2 or K562 as a model. I appreciate the authors' efforts in presenting many analyses and plots backing up each statement.

Comments on revised version:

In the second round of review, I think the authors have sufficiently addressed all of my previous comments. The study itself is very comprehensive, tackling all aspects of the HOT loci, though I still find the paper to be unnecessarily long and long-winded. That said, being consistent with the long and detailed paper, the provided Github repository and Zenodo archive is well-documented. I appreciate that the authors include detailed readme about the different datafiles available for readers. The list of HOT loci is probably the most useful asset in this manuscript and the authors did a good job documenting data availability in both Github and Zenodo.

---

## [Referee Report · Reviewer #3 (Public review)]

Summary:

Hudaiberdiev and Ovcharenko investigate regions within the genome where a high abundance of DNA associated proteins are located and identify DNA sequence feature enriched in these regions, their conservation in evolution, and variation in disease. Using ChIP-seq binding profiles of over 1,000 proteins in three human cell lines (HepG2, K562, and H1) as a data source they're able to identify nearly 44,000 high-occupancy target loci (HOT) that form at promoter and enhancer regions, thus suggesting these HOT loci regulate housekeeping and cell identity genes. Their primary investigative tool is HepG2 cells, but they employ K562 and H1 cells as tools to validate these assertions in other human cell types. Their analyses use RNA pol II signal, super enhancer, regular enhancer and epigentic marks to support the identification of these regions. The work is notable, in that it identifies a set of proteins that are invariantly associated with high-occupancy enhancers and promoters and argues for the integration of these molecules at different genomic loci. These observations are leveraged by the authors to argue HOT loci as potential sites of transcriptional condensates, a claim that they provide information in support of. Transcriptional condensates are an important "family" of condensates, regulating different types of genes and this work supports the hypothesis that they possess similar protein partner molecules as those thought to define such bodies.

---

## [Author Response]

The following is the authors’ response to the original reviews.

**Reviewer #1 (Public Review):**
Summary:This study explores the sequence characteristics and features of high-occupancy target (HOT) loci across the human genome. The computational analyses presented in this paper provide information into the correlation of TF binding and regulatory networks at HOT loci that were regarded as lacking sequence specificity.By leveraging hundreds of ChIP-seq datasets from the ENCODE Project to delineate HOT loci in HepG2, K562, and H1-hESC cells, the investigators identified the regulatory significance and participation in 3D chromatin interactions of HOT loci. Subsequent exploration focused on the interaction of DNA-associated proteins (DAPs) with HOT loci using computational models. The models established that the potential formation of HOT loci is likely embedded in their DNA sequences and is significantly influenced by GC contents. Further inquiry exposed contrasting roles of HOT loci in housekeeping and tissue-specific functions spanning various cell types, with distinctions between embryonic and differentiated states, including instances of polymorphic variability. The authors conclude with a speculative model that HOT loci serve as anchors where phase-separated transcriptional condensates form. The findings presented here open avenues for future research, encouraging more exploration of the functional implications of HOT loci.Strengths:The concept of using computational models to define characteristics of HOT loci is refreshing and allows researchers to take a different approach to identifying potential targets. The major strengths of the study lies in the very large number of datasets analyzed, with hundreds of ChIP-seq data sets for both HepG2 and K562 cells as part of the ENCODE project. Such quantitative power allowed the authors to delve deeply into HOT loci, which were previously thought to be artifacts.Weaknesses:While this study contributes to our knowledge of HOT loci, there are critical weaknesses that need to be addressed. There are questions on the validity of the assumptions made for certain analyses. The speculative nature of the proposed model involving transcriptional condensates needs either further validation or be toned down. Furthermore, some apparent contradictions exist among the main conclusions, and these either need to be better explained or corrected. Lastly, several figure panels could be better explained or described in the figure legends.

We thank the reviewer for their valuable comments.

- We have extended the study and included a new chapter focusing on the condensate hypothesis, added more supporting evidence (including the ones suggested by the reviewer), and made explicit statements on the speculative nature of this model.

- We have restructured the text to remove the sentences which might be construed as contradictory.

**Reviewer #2 (Public Review):**
Summary:The paper 'Sequence characteristic and an accurate model of abundant hyperactive loci in human genome' by Hydaiberdiev and Ovcharenko offers comprehensive analyses and insights about the 'high-occupancy target' (HOT) loci in the human genome. These are considered genomic regions that overlap with transcription factor binding sites. The authors provided very comprehensive analyses of the TF composition characteristics of these HOT loci. They showed that these HOT loci tend to overlap with annotated promoters and enhancers, GC-rich regions, open chromatin signals, and highly conserved regions, and that these loci are also enriched with potentially causal variants with different traits.Strengths:Overall, the HOT loci' definition is clear and the data of HOT regions across the genome can be a useful dataset for studies that use HepG2 or K562 as a model. I appreciate the authors' efforts in presenting many analyses and plots backing up each statement.Weaknesses:It is noteworthy that the HOT concept and their signature characteristics as being highly functional regions of the genome are not presented for the first time here. Additionally, I find the main manuscript, though very comprehensive, long-winded and can be put in a shorter, more digestible format without sacrificing scientific content.The introduction's mention of the blacklisted region can be rather misleading because when I read it, I was anticipating that we are uncovering new regulatory regions within the blacklisted region. However, the paper does not seem to address the question of whether the HOT regions overlap, if any, with the ENCODE blacklisted regions afterward. This plays into the central assessment that this manuscript is long-winded.The introduction also mentioned that HOT regions correspond to 'genomic regions that seemingly get bound by a large number of TFs with no apparent DNA sequence specificity' (this point of 'no sequence specificity' is reiterated in the discussion lines 485-486). However, later on in the paper, the authors also presented models such as convolutional neural networks that take in one-hot-encoded DNA sequence to predict HOT performed really well. It means that the sequence contexts with potential motifs can still play a role in forming the HOT loci. At the same time, lines 59-60 also cited studies that "detected putative drive motifs at the core segments of the HOT loci". The authors should edit the manuscript to clarify (or eradicate) contradictory statements.

We thank the reviewer for their valuable comments. Below are our responses to each paragraph in the given order:

We added a statement in the commenting and summarizing other publications that studied the functional aspects of HOT loci with the following sentence in the introduction part:

“Other studies have concluded that these regions are highly functionally consequential regions enriched in epigenetic signals of active regulatory elements such as histone modification regions and high chromatin accessibility”.

We significantly shortened the manuscript by (a) moving the detailed analyses of the computational model to the supplemental materials, and (b) shortening the discussions by around half, focusing on core analyses that would be most beneficial to the field.

Given that the ENCODE blacklisted regions are the regions that are recommended by the ENCODE guidelines to be avoided in mapping the ChIP-seq (and other NGS), we excluded them from our analyzed regions before mapping to the genome. Instead, we relied on the conclusions of other publications on HOT loci that the initial assessments of a fraction of HOT loci were the result of factoring in these loci which later were included in blacklisted regions.

We addressed the potential confusion by using the expression of “no sequence specificity” by (a) changing the sentence in the introduction by adding a clarification as “... with no apparent DNA sequence specificity in terms of detectible binding motifs of corresponding motifs” and (b) removing that part from the sentence in the discussions.

**Reviewer #3 (Public Review):**
Summary:Hudaiberdiev and Ovcharenko investigate regions within the genome where a high abundance of DNA-associated proteins are located and identify DNA sequence features enriched in these regions, their conservation in evolution, and variation in disease. Using ChIP-seq binding profiles of over 1,000 proteins in three human cell lines (HepG2, K562, and H1) as a data source they're able to identify nearly 44,000 high-occupancy target loci (HOT) that form at promoter and enhancer regions, thus suggesting these HOT loci regulate housekeeping and cell identity genes. Their primary investigative tool is HepG2 cells, but they employ K562 and H1 cells as tools to validate these assertions in other human cell types. Their analyses use RNA pol II signal, super-enhancer, regular-enhancer, and epigenetic marks to support the identification of these regions. The work is notable, in that it identifies a set of proteins that are invariantly associated with high-occupancy enhancers and promoters and argues for the integration of these molecules at different genomic loci. These observations are leveraged by the authors to argue HOT loci as potential sites of transcriptional condensates, a claim that they are well poised to provide information in support of. This work would benefit from refinement and some additional work to support the claims.Comments:(1) Condensates are thought to be scaffolded by one or more proteins or RNA molecules that are associated together to induce phase separation. The authors can readily provide from their analysis a check of whether HOT loci exist within different condensate compartments (or a marker for them). Generally, ChIPSeq signal from MED1 and Ronin (THAP11) would be anticipated to correspond with transcriptional condensates of different flavors, other coactivator proteins (e.g., BRD4), would be useful to include as well. Similarly, condensate scaffolding proteins of facultative and constitutive heterochromatin (HP1a and EZH2/1) would augment the authors' model by providing further evidence that HOT Loci occur at transcriptional condensates and not heterochromatin condensates. Sites of splicing might be informative as well, splicing condensates (or nuclear speckles) are scaffolded by SRRM/SON, which is probably not in their data set, but members of the serine arginine-rich splicing factor family of proteins can serve as a proxy-SRSF2 is the best studied of this set. This would provide a significant improvement to their proposed model and be expected since the authors note that these proteins occur at the enhancers and promoter regions of highly expressed genes.(2) It is curious that MAX is found to be highly enriched without its binding partner Myc, is Myc's signal simply lower in abundance, or is it absent from HOT loci? How could it be possible that a pair of proteins, which bind DNA as a heterodimer are found in HOT loci without invoking a condensate model to interpret the results?(3) Numerous studies have linked the physical properties of transcription factor proteins to their role in the genome. The authors here provide a limited analysis of the proteins found at different HOT-loci by employing go terms. Is there evidence for specific types of structural motifs, disordered motifs, or related properties of these proteins present in specific loci?(4) Condensates themselves possess different emergent properties, but it is a product of the proteins and RNAs that concentrate in them and not a result of any one specific function (condensates can have multiple functions!)(5) Transcriptional condensates serve as functional bodies. The notion the authors present in their discussion is not held by practitioners of condensate science, in that condensates exist to perform biochemical functions and are dissolved in response to satisfying that need, not that they serve simply as reservoirs of active molecules. For example, transcriptional condensates form at enhancers or promoters that concentrate factors involved in the activation and expression of that gene and are subsequently dissolved in response to a regulatory signal (in transcription this can be the nascently synthesized RNA itself or other factors). The association reactions driving the formation of active biochemical machinery within condensates are materially changed, as are the kinetics of assembly. It is unnecessary and inaccurate to qualify transcriptional condensates as depots for transcriptional machinery.1. This work has the potential to advance the field forward by providing a detailed perspective on what proteins are located in what regions of the genome. Publication of this information alongside the manuscript would advance the field materially.

We thank the reviewer for constructive comments and suggestions. Below are our point-by-point responses:

(1) We added a new short section “Transcriptional condensates as a model for explaining the HOT regions” with additional support for the condensate hypothesis, wherein some of the points raised here were addressed. Specifically, we used a curated LLPS proteins (CD-CODE) database and provided statistics of those annotation condensate-related DAPs.

Regarding the DAPs mentioned in this question, we observed that the distributions corresponding ChIP-seq peaks confirm the patterns expected by the reviewer (Author response image 1). Namely:

- MED1 and Ronin (THAP11) are abundant in the HOT loci, being present 67% and 64% of HOT loci respectively.

- While the BRD4 is present in 28% of the HOT loci, we observed that the DAPs with annotated LLPS activity ranged from 3% to 73%, providing further support for the condensate hypothesis.

- ENCODE database does not contain ChIP-seq dataset for HP1A. EZH2 peaks were absent in the HOT loci (0.4% overlap), suggesting the lack of heterochromatin condensate involvement.

- Serine-rich splicing factor family proteins were present only in 7.7% of the HOT loci, suggesting the absence or limited overlap with splicing condensates or nuclear speckles.

**Author response image 1. sa4fig1:** 

(2) In this study we selected the TF ChIP-seq datasets with stringent quality metrics, excluding those which had attached audit warning and errors. As a result, the set of DAPs analyzed in HepG2 did not include MYC, since the corresponding ChIP-seq dataset had the audit warning tags of "borderline replicate concordance, insufficient read length, insufficient read depth, extremely low read depth". Analyses in K562 and H1 did include MYC (alongside MAX) ChIP-seq dataset.

To address this question, we added the mentioned ChIP-seq dataset (ENCODE ID: ENCFF800JFG) and analyzed the colocalization patterns of MYC and MAX. We observed that the MYC ChIP-seq peaks in HepG2 display spurious results, overlapping with only 5% of HOT loci. Meanwhile in K562 and H1, MYC and MAX are jointly present in 54% and 44% of the HOT loci, respectively (Author response image 2).

**Author response image 2. sa4fig2:** 

These observations were also supported by Jaccard indices between the MYC and MAX ChIP-seq peaks. To do this analysis, we calculated the pairwise Jaccard indices between MYC and MAX and divided them by the average Jaccard indices of 2000 randomly selected DAP pairs. In K562 and H1, the Jaccard indices between MYC and MAX are 5.72x and 2.53x greater than the random background, respectively. For HepG2, the ratio was 0.21x, clearly indicating that HepG2 MYC ChIP-seq dataset is likely erroneous.

**Author response image 3. sa4fig3:** 

(3) Despite numerous publications focusing on different structural domains in transcription factors, we could not find an extensive database or a survey study focusing on annotations of structural motifs in human TFs. Therefore, surveying such a scale would be outside of this study’s scope. We added only the analysis of intrinsically disordered regions, as it pertains to the condensate hypothesis. To emphasize this shortcoming, we added the following sentence to the end of the discussions section.

“Further, one of the hallmarks of LLPS proteins that have been associated with their abilities to phase-separate is the overrepresentation of certain structural motifs, which we did not pursue due to size limitations.”

(4, 5) We agree with these statements and thank the reviewer for pointing out this faulty statement. We modified the sections in the discussions related to the condensates and removed the part where we implied that the condensate model could be because of mostly a single function of TF reservoir.

(6) We added a table to the supplemental materials (Zenodo repository) with detailed annotation of HOT and non-HOT DAP-bound loci in the genome.

**Recommendations for the authors:**

**Reviewing Editor (Recommendations For The Authors):**
The clause with "inadequate" would be dropped if the authors sufficiently address reviewer concerns about clarity of writing, including:(1) Editing the title to better reflect the findings of the paper.(2) Making clear that the condensate model is speculative and not explicitly tested in this study (and may be better described as a hypothesis).(3) Resolving apparent contradictions regarding DNA sequence specificity and the interpretation of ChIP-seq signal intensity.(4) Better specifying and justifying model parameters, thresholds, and assumptions.(5) Shortening the manuscript to emphasize the main, well-supported claims and to enhance readability (especially the discussion section).

We thank the Editor for their work. We followed their advice and implemented changes and additions to address all 5 points.

**Reviewer #1 (Recommendations For The Authors):**
(1) The title "Sequence characteristics and an accurate model of abundant hyperactive loci in the human genome" does not accurately reflect the findings of the paper. We are unclear as to what the 'accurate model' refers to. Is it the proposed model 'based on the existence of large transcriptional condensates' (abstract)? If so, there are concerns below regarding this statement (see comment 2). If the authors are referring to the computational modeling presented in Figure 5, it is unclear that any one of them performed that much better than the others and the best single model was not identified. Furthermore, the models being developed in the study constitute only a portion of the paper and lacked validation through additional datasets. Additionally, sequence characteristics were not a primary focus of the study. Only figure 5 talks about the model and sequence characteristics, the rest of the figures are left out of the equation.

We agree with and thank the reviewer for this idea of clarifying the intended meaning.

(1) We changed the title and clarified that the computational model is meant:

“Functional characteristics and a computational model of abundant hyperactive loci in the human genome”.

(2) Shortened the part of the manuscript discussing the computational models and pointed out the CNNs as “the best single model”.

(2) The abstract and discussion (and perhaps the title) propose a model of transcriptional condensates in relation to HOT loci. However, there is no data provided in the manuscript that relates to condensates. Therefore, anything relating to condensates is primarily speculative. This distinction needs to be properly made, especially in the abstract (and cannot be included in the title). Otherwise, these statements are misleading. Although the field of transcriptional condensates is relatively new, there have been several factors studied. The authors could include in Figure 2d which factors have been shown to form transcriptional condensates. This might provide some support for the model, though it would still largely remain speculative unless further testing is done.

We added a new short chapter “Transcriptional condensates as a model for explaining the HOT regions”, with additional analyses testing the condensates hypothesis. We provided supportive evidence by analyzing the metrics used as hallmarks of condensates including the distributions of annotated condensate-related proteins, nascent transcription, and protein-RNA interaction levels in HOT loci. Still, we acknowledge that this is a speculative hypothesis and we clarified that with the following statement in the discussions:

“It is important to note here that our proposed condensate model is a speculative hypothesis. Further experimental studies in the field are needed to confirm or reject it.”

(3) Several apparent contradictions exist throughout the manuscript. For example, "HOT locus formation are likely encoded in their DNA sequences" (lines 329-330) vs the proposed model of formation through condensates (abstract). These two statements do not seem compatible, or at the very least, the authors can explain how they are consistent with each other. Another example: "ChIP-seq signal intensity as a proxy for... binding affinity" (line 229) vs. "ChIP-seq signal intensities do not seem to be a function of the DNA-binding properties of the DAPs" (lines 259-260). The first statement is the assumption for subsequent analyses, which has its own concerns (see comment 4). But the conclusion from that analysis seems to contradict the assumption, at least as it is stated.

In this study, we argue that the two statements may not necessarily contradict each other. We aimed to (a) demonstrate that the observed intensity of DAP-DNA interactions as measured by ChIP-seq experiments at HOT loci cannot be explained with direct DNA-binding events of the DAPs alone and (b) propose a hypothesis that this observation can be at least partially explained if the HOT loci have the propensity to either facilitate or take part in the formation of transcriptional condensates.

One of the conditions for condensates to form at enhancers was shown to be the presence of strong binding sites of key TFs (Shrinivas et al. 2019 “Enhancer features that drive the formation of transcriptional condensates”), where the study was conducted using only one TF (OCT4) and one coactivator (MED1). To the best of our knowledge, no such study has been conducted involving many TFs and cofactors simultaneously. We also know that the factors that lead to liquid-to-liquid phase separation include weak multivalent IDR-IDR, IDR-DNA, and IDR-RNA interactions. As a result, the observed total sum of ChIP-seq peaks in HOT loci is the direct DNA-binding events combined with the indirect DAP-DNA interactions, some of which may be facilitated by condensates. And, the fact that CNNs can recognize the HOT loci with high accuracy suggests that there must be an underlying motif grammar specific to HOT loci.

We emphasized this conclusion in the discussions.

The comment on using the ChIP-seq signal as a proxy for DNA-binding affinity is addressed under comment 4.

(4) In lines 229-230, the authors used "the ChIP-seq signal intensity as a proxy for the DAP binding affinity." What is the basis for this assumption? If there is a study that can be referenced, it should be added. However, ChIP-seq signal intensity is generally regarded as a combination of abundance, frequency, or percentage of cells with binding. RNA Pol2 is a good example of this as it has no specific binding affinity but the peak heights indicate level of expression. Therefore, the analyses and conclusions in Figure 4, particularly panel A, are problematic. In addition, clarification from lines 258-260 is needed as it contradicts the earlier premise of the section (see comment 3).

We thank the reviewer for pointing out this error. The main conclusion of the paragraph is that the average ChIP-seq signal values at HOT loci do not correlate well with the sequence-specificity of TFs. We reworded the paragraph stating that we are analyzing the patterns of ChIP-seq signals across the HOT loci, removing the part that we use them as a proxy for sequence-specific binding affinity.

(5) In Figure 1A, the authors show that "the distribution of the number of loci is not multimodal, but rather follows a uniform spectrum, and thus, this definition of HOT loci is ad-hoc" (lines 92-95). The threshold to determine how a locus is considered to be HOT is unclear. How did the authors decide to use the current threshold given the uniform spectrum observed? How does this method of calling HOT loci compare to previous studies? How much overlap is there in the HOT loci in this study versus previous ones?

We moved the corresponding explanation from the supplemental methods to the main methods section of the manuscript.

Briefly, our reasoning was as follows: assuming that an average TFBS is 8bp long and given that we analyze the loci of length 400bp, we can set the theoretical maximum number of simultaneous binding events to be 50. Hence, if there are >50 TF ChIP-seq peaks in a given 400bp locus, it is highly unlikely that the majority of ChIP-seq peaks can be explained by direct TF-DNA interactions. The condition of >50 TFs corresponded to the last four bins of our binning scale, which was used as an operational definition for HOT loci.

We have compared our definition of HOT loci to those reported in previous studies by Remaker et al. and Boyle et al. The results of our analyses are in lines 147-154.

(6) In Figure 3B, the authors state that of "the loop anchor regions with >3 overlapping loops, 51% contained at least one HOT locus, suggesting an interplay between chromatin loops and HOT loci." However, it is unclear how "51%" is calculated from the figure. Similarly, in the following sentence, "94% of HOT loci are located in regions with at least one chromatin interaction". It is unclear as to how the number was obtained based on the referenced figure.

Initially, the x-axis on the Figure 3B was missing, making it hard to understand what we meant. We added the x-axis numbers and changed the “51%” to “more than half”. We intend to say that, of the loci with 4 and 5 overlapping loops, exactly 50% contain at least one HOT locus. However, since for x=6 the percentage is 100% (since there’s only one such locus), the percentage is technically “more than half”.

The percentage of HOT loci engaging in chromatin interaction regions (91%) was calculated by simply overlapping the HOT regions with Hi-C long-range contact anchors. The details of extracting these regions using FitHiChip are described in Supplemental Methods 1.3.

(7) While we have a limited basis to evaluate computational models, we would like to see a clearer explanation of the model set-up in terms of the number of trained vs. test datasets. In addition, it would be interesting to see if the models can be applied to data from different cell lines.

We added the table with the sizes of the datasets used for classification in Supplemental Methods 1.6.1.

Evaluating the models trained on the HOT loci of HepG2 and K562 on other cell lines would pose challenges since the number of available ENCODE TF ChIP-seq datasets is significantly less compared to the mentioned cell lines. Therefore, we conducted the proposed analysis between the studied cell lines. Specifically, we used the CNN models trained on HOT and regular enhancers of HepG2 and K562. Then, we evaluated each model on the test sets of each classification experiment (Author response image 4). We observed that the classification results of the HOT loci demonstrated a higher level of tissue-specificity compared to the same classification results of the regular enhancers.

**Author response image 4. sa4fig4:** 

(8) Lines 349-351. The significance of highly expressed genes being more prone to having multiple HOT loci, and vice versa, appears conventional and remains unclear. Intuitively, it makes sense for higher expressed genes to have more of the transcriptional machinery bound, and would bias the analysis. One way to circumvent this is to only analyze sequence-specific TFs and remove ones that are directly related to transcription machinery.

We thank the reviewer for this suggestion. Our attempt to re-annotate the HOT loci with only sequence-specific TFs led to a significantly different set of loci, which would not be strictly comparable to the HOT loci defined by this study. Analyzing these new sets of loci would create a noticeable departure from the flow of the manuscript and further extend the already long scope of the study.

Moreover, numerous studies have shown that super-enhancers recruit large numbers of TFs via transcriptional condensates (Boija et al., 2018; Cho et al., 2018; Sabari et al., 2018). We hope that our results can serve as data-driven supportive evidence for those studies.

(9) Lines 393-396. We would like to see a reference to the models shown in the figures, if these models have been published previously.

We could not understand the question. The lines 393-396 contains the following sentence:

“However, many of the features of the loci that we’ve analyzed so far demonstrated similar patterns (GC contents, target gene expressions, ChIP-seq signal values etc.) when compared to the DAP-bound loci in HepG2 and K562, suggesting that albeit limited, the distribution of the DAPs in H1 likely reflects the true distribution of HOT loci.”

In case the question was about the models that we trained to classify the HOT loci, we included the models and codebase to Zenodo and GitHub repository.

(10) Values in Figure 7D are not reflected in the text. Specifically, the text states "Average ... phastCons of the developmental HOT loci are 1.3x higher than K562 and HepG2 HOT loci (Figure 7D)" (lines 408-409). Figure 7D shows conservation scores between HOT enhancers vs promoters for each cell line, and does not seem to reflect the text.

We modified the figure to reflect the statement appropriately.

(11) Methodology should include a justification for the use of the Mann-Whitney U-test (non-parametric) over other statistical tests.

We added the following description to the methods section:

“For calculating the statistical significance, we used the non-parametric Mann-Whitney U-test when the compared data points are non-linearly correlated and multi-modal. When the data distributions are bell-curve shaped, the Student’s t-test was used.“

Minor:(1) Figure 2b was never mentioned in the paper. This can be added alongside Figure S6C, line 148.

Indeed, Figure 2B was supposed to be listed together with Figure S6C, which was omitted by mistake. It was corrected.

(2) Supplementary Figure 8 has two Cs. Needs to be corrected to D.

Fixed.

(3) Figure 3B is missing labels on the x-axis.

Fixed.

(4) The horizontal bar graph on the bottom left of Figure 1E needs to be described in the figure legend.

Description added to the figure caption.

(5) Line 345, Fig 15A should be Fig S15A.

Corrected.

**Reviewer #2 (Recommendations For The Authors):**
I listed all my concerns about the paper in the public comments. I think the manuscript is very comprehensive and it is valuable, but it should be cut short and presented in a more digestible way.

We thank the reviewer for their valuable comments and suggestions. We addressed all the concerns listed in the public comments. We shortened the manuscript by reducing the paragraph that focuses on computational classification models and reduced the discussions by about half in length.

Line 55: What are chromatin-associated proteins, i.e. are they histone modifications?

To clarify the definition used from the citation we changed the sentence to the following:

“For instance, Partridge et al. studied the HOT loci in the context of 208 proteins including TFs, cofactors, and chromatin regulators which they called chromatin-associated proteins.”

Though most of the paper can be cut short to avoid analysis paralysis for readers, there are details that still need filling in. For example, how did the authors perform PCA analysis, i.e. what are the features of each data point in the PCA analysis? Lines 214-215: How do we calculate the number of multi-way contacts in Hi-C data?

We added clarifying descriptions and changed the mentioned sentences to the following:

PCA:

“To analyze the signatures of unique DAPs in HOT loci, we performed a PCA analysis where each HOT locus is represented by a binary (presence/absence) vector of length equal to the total number of DAPs analyzed.”

Multi-way contacts on loop anchors:

“To investigate further, we analyzed the loop anchor regions harboring HOT loci and observed that the number of multi-way contacts on loop anchors (i.e. loci which serve as anchors to multiple loops) correlates with the number of bound DAPs (rho=0.84 p-value<10E-4; Pearson correlation). “

- Lines 251-252: How did the referenced study categorize DAPs? It is important for any manuscript to be self-contained.

We added the explanation and changed the sentence to the following:

“To test this hypothesis, we classified the DAPs into those two categories using the definitions provided in the study (Lambert et al. 2018) 28, where the TFs are classified by manual curation through extensive literature review and supported by annotations such as the presence of DNA-binding domains and validated binding motifs**.** Based on this classification, we categorized the ChIP-seq signal values into these two groups.“

- Lines 181-185, sentences starting with 'To test' can be moved to the methods, leaving only brief mentions of the statistic tests if needed.

We removed the mentioned sentence and moved to the supplemental methods (1.4).

- Lines 217-220: I find this sentence extremely redundant unless it can offer more specific insights about a particular set of DAPs or if the DAPs are closer/or a proven distal enhancer to a confirmed causal gene.

We removed the mentioned sentence from the text.

- Lines 243-246: How did the authors determine the set DAPs that have stabilizing effects, and how exactly are the 'stabilizing effects' observed/measured?

We added explanations to Supplemental Methods 3.1 and Fig S18, S19.

While addressing this comment we realized that the reported value of the ratio is 1.91x, not 1.7x. We corrected that value in the main text and added the p-value.

- When discussing the phastCons scores analyses, such as in lines 268-271, how did the authors calculate the relationship between phastCons scores and HOT loci, i.e. was the score averaged across the 400-bp locus to obtain a locus-specific conservation score?

Yes, per-locus conservation scores were averaged over the bps of loci. We added this clarification to the methods.

- Line 311: What is the role of the 'control sets' in the analyses of the sequence's relationship with HOT?

In this specific case, the control sets are used as background or negative sets to set up the classification tasks. In other words, we are asking, whether the HOT loci can be distinguished when compared to random chromatin-accessible regions, promoters, or regular enhancers. We clarified this in the text.

- I also find the discussion about different machine learning methods that classify HOT loci based on sequence contexts quite redundant UNLESS the authors decide to go further into the features' importance (such as motifs) in the models that predict/ are associated with HOT loci, which in itself can constitute another study.

We agree with the reviewer, and shortened the part with the discussions of models by limiting it to only 3 main models and moved the rest to the supplemental materials.

- Can the authors clarify where they obtain data on super-enhancers?

We obtained the super-enhancer definitions from the original study (Hnisz et al. 2013, PMID: 24119843) where the super-enhancers were defined for multiple cell lines. We clarified this in the methods.

- Figure 1B, the x and y axis should be clarified.

We clarified it by using MAX as an example case in the figure caption as follows:

“Prevalence of DAPs in HOT loci. Each dot represents a DAP. X-axis: percentage of HOT loci in which DAP is present (e.g. MAX is present in 80% of HOT loci). Y-axis: percentage of total peaks of DAPs that are located in HOT loci (e.g. 45% of all the ChIP-seq peaks of MAX is located in the HOT loci). Dot color and size are proportional to the total number of ChIP-seq peaks of DAP.”

**Reviewer #3 (Recommendations For The Authors):**
The list of proteins associated with different types of genomic loci at a meta level (enhancers, promoters, and gene body etc.), and an annotation of the genome at the specific loci level.The authors use a wide range of acronyms throughout the text and figure legends, they do a reasonably good job, but the main text section "HOT-loci are enriched in causal variants" and Figure 8 would be materially improved if they held it to the same standard.Size is a physical property and not a physicochemical property.

We thank the reviewer for their comments and suggestions. We added a table to supplemental files with detailed annotations of analyzed loci.

We reviewed the section “HOT loci are enriched in causal variants” and corrected a few mismatches in the acronyms.